# Dynamic expression of tRNA-derived small RNAs define cellular states

Srikar Krishna[1,2,3,†], Daniel GR Yim[4,†] (iD), Vairavan Lakshmanan[2,3,†], Varsha Tirumalai[3,5], Judice LY Koh[4], Jung Eun Park[6], Jit Kong Cheong[7] (iD), Joo Leng Low[4], Michelle JS Lim[4] (iD), Siu Kwan Sze[6] (iD), Padubidri Shivaprasad[5], Akash Gulyani[2], Srikala Raghavan[1,*] (iD), Dasaradhi Palakodeti[2,**] (iD) & Ramanuj DasGupta[4,***] (iD)

## Abstract

Transfer RNA (tRNA)-derived small RNAs (tsRNAs) have recently emerged as important regulators of protein translation and shown to have diverse biological functions. However, the underlying cellular and molecular mechanisms of tsRNA function in the context of dynamic cell-state transitions remain unclear. Expression analysis of tsRNAs in distinct heterologous cell and tissue models of stem vs. differentiated states revealed a differentiation-dependent enrichment of 5′-tsRNAs. We report the identification of a set of 5′-tsRNAs that is upregulated in differentiating mouse embryonic stem cells (mESCs). Notably, interactome studies with differentially enriched 5′-tsRNAs revealed a switch in their association with "effector" RNPs and "target" mRNAs in different cell states. We demonstrate that specific 5′-tsRNAs can preferentially interact with the RNA-binding protein, Igf2bp1, in the RA-induced differentiated state. This association influences the transcript stability and thereby translation of the pluripotency-promoting factor, c-Myc, thus providing a mechanistic basis for how 5′-tsRNAs can modulate stem cell states in mESCs. Together our study highlights the role of 5′-tsRNAs in defining distinct cell states.

**Keywords** c-Myc; Igf2bp1; stem cell differentiation; translation; tRNA-derived small RNAs (tsRNAs)

**Subject Categories** Development & Differentiation; RNA Biology; Stem Cells

## Introduction

Cell differentiation is a highly dynamic process involving the spatio-temporal expression of proteins that control specific cellular phenotypes. Elucidating the molecular mechanisms underlying cell-state transitions is important in understanding the process of differentiation. Gene expression is tightly regulated at the transcriptional, post-transcriptional, and translational levels. Multiple studies have identified the function of miRNAs in modulating gene expression to drive pluripotency and differentiation [1]. However, miRNAs represent only a subset of non-coding RNAs (ncRNAs). The role of other small ncRNAs as modulators of cell-state transition and differentiation has remained relatively under-explored.

More recently, high-throughput sequencing of various biological samples has led to the discovery of a novel species of small RNAs called the transfer RNA (tRNA)-derived small RNAs. Based on the region of tRNA from which these small RNAs are processed, these can be classified into five different species: 32–34 nt 5′- and 3′-halves; 18–22 nt 5′- and 3′-tRNA fragments (tRFs); and 3′U tRFs [2,3]. Endoribonucleases RNase Z, Dicer, and angiogenin have been implicated in the processing of the different tRNA-derived small RNA species [4–6]. The widespread and conserved expression of these small RNAs across several biological systems has piqued the interest in understanding the functional significance of these tRNA-derived small RNAs. There is growing evidence that these small RNAs are involved in regulating several biological processes, such as tumor suppression [7], ribosomal biogenesis in cancer [8], paternal epigenetic inheritance [9,10], and LTR-retrotransposon control [11] through distinct mechanisms. One of the highly conserved and abundant species is the 5′-halves, tiRNAs, or tsRNAs (henceforth

1 Centre for Inflammation and Tissue Homeostasis, Institute for Stem Cell Science and Regenerative Medicine, Bangalore, India
2 Technologies for the Advancement of Science, Institute for Stem Cell Science and Regenerative Medicine, Bangalore, India
3 SASTRA University, Thirumalaisamudram, Thanjavur, India
4 Precision Oncology, Genome Institute of Singapore, Singapore City, Singapore
5 National Centre for Biological Sciences, Bangalore, India
6 School of Biological Sciences, Nanyang Technological University, Singapore City, Singapore
7 Program in Cancer and Stem Cell Biology, Duke-NUS Medical School, Singapore City, Singapore
*Corresponding author. Tel: +91 80 23666743; E-mail: srikala@instem.res.in
**Corresponding author. Tel: +91 80 23666767; E-mail: dasaradhip@instem.res.in
***Corresponding author. Tel: +65 6808 8005; E-mail: dasguptar@gis.a-star.edu.sg
†These authors are contributed equally to this work
[Correction added on 20 June 2019, after first online publication: the authors' affiliations have been corrected.]

called 5′ tRNA-derived small RNA or 5′-tsRNA). 5′-tsRNAs were initially identified as regulators of *Tetrahymena* stress response [5,6,12]. While several reports have shown that these species of small RNAs are produced in response to various stress conditions [6,13], it has also been suggested that the involvement of certain hormonal pathways can initiate the production of 5′-halves [14]. Functional studies have revealed a role for these angiogenin-cleaved 5′-tsRNAs in repressing translation initiation [6,15,16]; however, it remains unclear whether they are global translational repressors or exhibit specificity in their action. Additionally, while 5′-tsRNAs have been shown to interact with several RNA-binding proteins (RBPs) [7,15], the functional consequence of such interactions has remained largely unexplored.

Here, we report differential enrichment of 5′-tsRNAs in a broad range of heterologous differentiation-associated cell states. Specifically, our study focuses on 5′-tsRNAs enriched during retinoic acid (RA)-induced differentiation of mESCs. Functional characterization using antisense oligo (ASO)-mediated knockdown studies suggests that 5′-tsRNAs can modulate mESC differentiation by regulating active translation in addition to their previously reported role in translation initiation [15]. RNA and proteome interaction studies with candidate 5′-tsRNAs, specifically tsGlnCTG, tsGluTTC, tsGlyGCC, and tsValCAC, revealed that individual 5′-tsRNAs have both unique and overlapping interactomes. Mechanistically, we show that tsGlnCTG, via its interaction with IGF2BP1, regulates both the transcript stability and translation of c-Myc. Altogether, our data implicate a previously unknown function for 5′-tsRNAs in defining cell states by regulating the expression and function of tsRNA-interacting mRNAs and proteins.

## Results

### Small RNA-seq in heterologous cell systems reveals that the expression of 5′-tsRNAs is consistently enriched in differentiated states compared with isogenic stem-like state

Deep sequencing of small RNAs in mESCs cultured under different conditions including Wnt3a+LIF (leukemia inhibitory factor; enhanced pluripotency), LIF alone (baseline pluripotency), and retinoic acid (RA; differentiation) in biological replicates (Fig EV1A) revealed a distinct population of 30–35 nt species (Figs 1A and EV1B) enriched during differentiation. Intriguingly, a large proportion of these 30–35 nt species mapped to tRNAs (Figs 1B and EV1C), and they represent one of the largest pool of small RNAs in differentiating states (Fig 1C). Specifically, in RA-treated differentiating mESCs 81.4% of these reads mapped to tRNAs, which dropped to 72% in LIF and to 50.3% in Wnt3a+LIF culture conditions (Fig 1B and Dataset EV1). These results suggest that the enrichment of tsRNAs inversely correlates with the pluripotent state of mESCs. Notably, most tsRNAs were derived from the 5′-halves (hence 5′-tsRNAs), starting from nucleotide 1–4 of parent tRNAs and terminating prior to the anticodon loop (Fig 1D, D′ and E), suggesting specific processing of tRNAs. It is intriguing to note that of the 468 tRNA loci that could potentially be processed into 5′-tsRNAs, only a few are selected for processing (Figs 1D′ and EV1D). Additionally, there did not seem to be a correlation between the tRNA gene copy number and the abundance of 5′-tsRNAs (Fig EV1D). As can be seen

CysGCA and AlaAGC have a high/comparable copy number, but are not selected for processing into tsRNA. Similarly, MetCAT that has a copy number comparable to GluTTC (one of the highly represented tsRNA) does not produce tsRNAs. In contrast, GlyCCC which has a relatively low copy number produces high abundance of tsRNAs, suggesting that is little or no correlation between the tRNA gene copy number and the abundance of tsRNAs (Fig EV1D). Previous reports have suggested the involvement of angiogenin in the processing of these 5′-tsRNAs [5,6,17,18]; however, this processing yields a 2′,3′ cyclic phosphate at the 3′ end. These fragments would be missed in the conventional small RNA library preparation due to the inability of the adapters to ligate to these fragments. To ascertain whether the 5′-tsRNAs observed in our system are angiogenin-dependent, we treated the RNA isolated from 48 h of RA-induced differentiating cells with T4 polynucleotide kinase and sequenced the libraries. Interestingly, we only observed a modest 15.3% increase in the reads mapping to the tRNA (Fig EV1E), suggesting the possible involvement of other enzymes in the cleavage of tRNAs into 5′-tsRNAs at this stage.

To investigate whether the phenomenon of tsRNA enrichment could function as a general read out of cell-state transitions, we performed small RNA sequencing on a variety of well-defined heterologous models for stem vs. differentiated states. Specifically, we identified 5′-tsRNAs from sorted stem and differentiating cell populations from murine skin [CD34$^+$α6$^+$ stem vs. CD34$^-$α6$^+$ non-stem cells (Fig EV1F); human mammary epithelial cells (HMECs) at different stages of oncogenic transformation as they gain stemness (Fig EV1G and G′) [19,20]; and CD44$^+$CD24$^-$ (stem) vs. CD24$^+$ (non-stem) cells in two breast cancer cell lines, namely MDA-MB-231 (Fig EV1H) and HS578T (Fig EV1I)]. Remarkably, we found a consistent enrichment of the 5′-tsRNAs in all the differentiated cell states compared with the various matched isogenic stem-like states (Fig EV1J). Collectively, our data suggest that high levels of 5′-tsRNA expression can serve as a global marker for stem vs. differentiated states.

### Dynamic expression of a group of 5′-tsRNAs during mESC differentiation

To understand the effect of 5′-tsRNAs in defining cell states, we decided to focus our functional studies using the mESC model. Our analysis revealed that only a specific population of tRNAs are selected for processing into 5′-tsRNA (Figs 1D′ and 2A, and EV1D). The 5′-tsRNAs predominantly corresponded to GlnCTG, GlyGCC, GluTTC, LysTTT, and ValCAC/AAC tRNAs (Fig 2A and Dataset EV2) across the three stem states tested. We further validated the differential enrichments observed through sequencing by small RNA qPCR (Fig EV2A and A′). We performed Northern hybridization to compare the expression levels of the five candidate 5′-tsRNAs between LIF- and RA-treated mESCs at 12, 24, 48, and 144 h (Figs 2B, C and C′, and EV2B). In comparison with LIF, RA-treated mESCs showed reduced levels of all tested 5′-tsRNAs after 12 h and 24 h and a subsequent increase in its expression at ~ 48 h (Figs 2B, C and C′, and EV2B). While levels of most 5′-tsRNAs showed a sharp increase at 144 h after RA treatment, tsGlnCTG and tsGluTTC levels saturated at 48 h of RA treatment (Figs 2B, C and C′, and EV2B). Importantly, the change in the levels of 5′-tsRNAs was not reflected on the parent tRNA levels (Figs 2B, C′, and EV2B and C), suggesting that levels of 5′-tsRNAs were not a function of

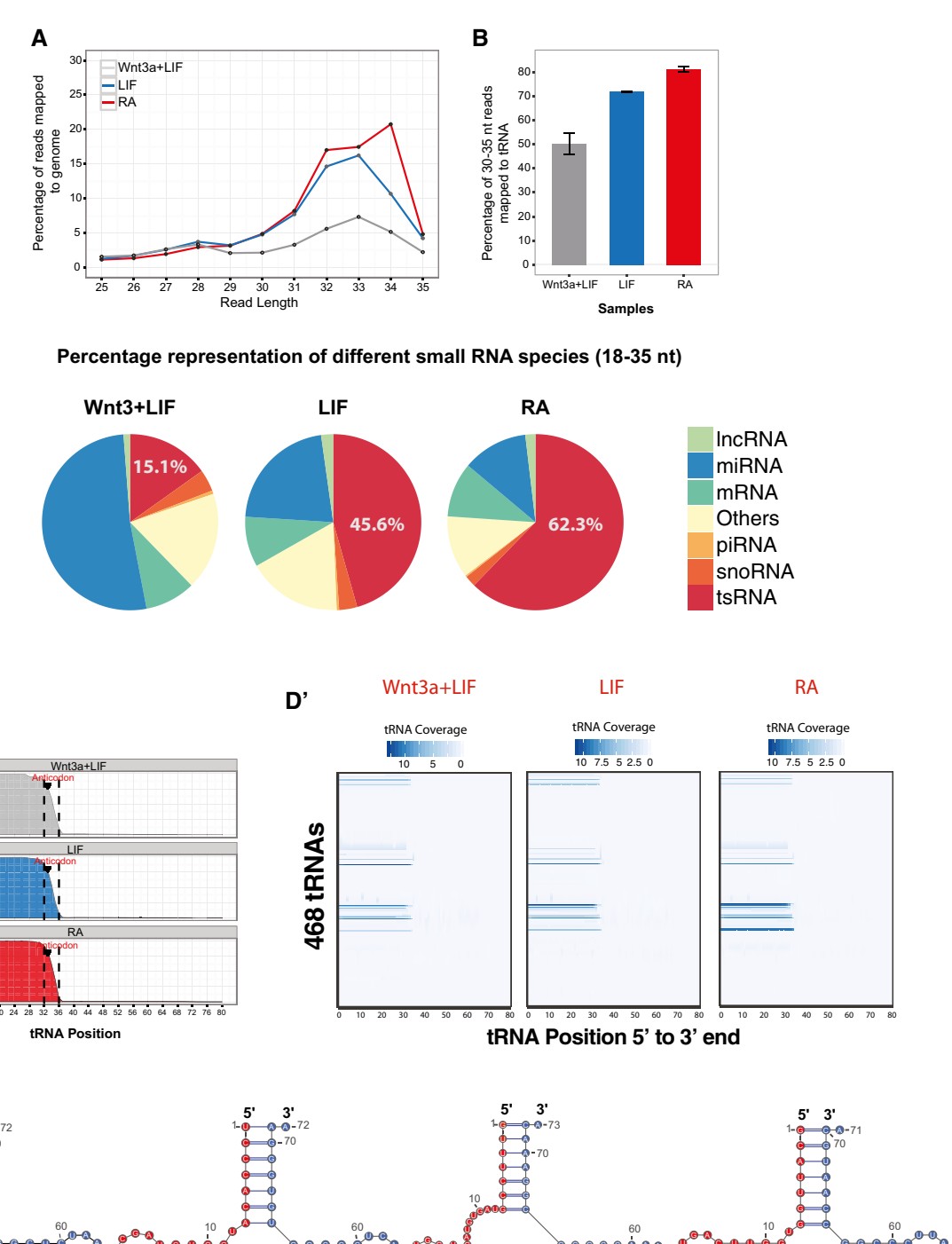

**Figure 1.**

◀

**Figure 1. Small RNA profiling of RA-induced differentiating mESCs.**

A   Distribution of 25–35 mer reads mapping to the mouse genome.
B   Proportion of 30–35 nt genomic reads matching mouse tRNA sequences. The sequencing was done in duplicates. Error bars represent SEM, and significance was calculated using unpaired *t*-test.
C   Pie charts depicting the different small RNA populations sequenced in the three stem cell states.
D   Per base sequence coverage of 30–35 nt reads across parental tRNA sequence. Anticodon positions are demarcated by discontinuous lines (blue = LIF-treated, gray = Wnt3 + LIF-treated, red = RA-treated mESCs).
D'  Per base coverage of 30–35 nt reads across the 468 individual parental tRNAs plotted as heat maps.
E   Positional heat map of reads mapped to tRNA$^{GlnCTG}$, tRNA$^{GluTTC}$, tRNA$^{ValAAC}$, and tRNA$^{GlyGCC}$ residues. The yellow box indicates the anticodon region.

differential expression of the parental tRNAs and that only a minor population of the tRNAs may be processed into 5′-tsRNAs.

## Differentially expressed 5′-tsRNAs modulate stem cell pluripotency in mESCs

To understand the effect of 5′-tsRNAs in defining cell states, we performed functional studies either by transfecting tsRNA mimics in LIF conditions or by blocking tsRNA function with antisense oligos (ASOs) in RA-induced differentiating conditions. First, we performed alkaline phosphatase (AP) assays as a measure of pluripotency to study the effect of increased tsRNA mimic expression (Fig EV3A) under stem conditions. Transfection of tsRNA mimics collectively (tsGlnCTG, tsGluTTC, tsLysCTT/TTT) or individually in LIF-treated stem-like growth conditions led to a mild decrease in AP activity (Fig EV3B). Moreover, none of the stemness-related markers changed in response to the increased tsRNA levels under LIF conditions (Fig EV3C). Concurrently, we observed minimal changes in the transcriptome when 5′-tsRNAs mimics were transfected under LIF conditions, suggesting a minor role for these small RNAs in the stem state (Fig EV3D).

Conversely, blocking 5′-tsRNA function using a pool of antisense oligos (against tsGlnCTG, tsGlyGCC, tsGlyCCC, tsGluTTC, tsLysTTT during RA-induced differentiation; Fig EV3E) led to a modest increase in AP activity (Fig 3A). Corroborating this initial observation, some of the stemness markers such as *Klf4* and *c-Myc* that are downstream of the JAK-STAT pathway were also found to be upregulated upon ASO-mediated inhibition of 5′-tsRNAs under differentiating conditions (Figs 3B and EV3F). Whole transcriptome profiling revealed significant alterations in the expression of several transcripts (Fig 3C and Dataset EV3). KEGG pathway analysis of the upregulated transcript showed a significant enrichment of genes that regulate stem cell pluripotency (Figs 3D and EV3F). Interestingly, *Lif* and *Wnt3*, a combination that drives the cells toward a more homogeneous stem state, were found to be upregulated in our transcriptome data (Figs 3E and EV3F, and Dataset EV3). Taken together, our data suggest that the primary role of 5′-tsRNAs may be to restrain the expression of stemness-promoting genes, thereby facilitating differentiation.

## 5′-tsRNAs interact with a specific set of proteins and transcripts during mESC differentiation

Recent studies have highlighted a crucial role for tsRNA–protein interactions, in regulating biological processes [7,15,21,22]. Given our observation that 5′-tsRNAs expression may define cell states, we reasoned that characterizing the proteins and mRNAs that interact

with 5′-tsRNAs would provide a molecular basis for their function. As the expression of 5′-tsRNAs was synchronously high at 48 h of RA treatment, we performed pulldown assays followed by mass spectrometry using four different tsRNA mimics (tsGlnCTG, tsGlyGCC, tsGluTTC, and tsValCCC) at this time point. Analysis of the protein interactome revealed that the four 5′-tsRNAs interacted with a common set of 109 proteins (Fig 4A and Dataset EV4), suggesting that these 5′-tsRNAs may exhibit a similar mode of action. It is important to note that a minor proportion of proteins are exclusive to specific 5′-tsRNAs, highlighting the possibility for unique mechanisms associated with individual 5′-tsRNAs (Fig 4A). GO annotation of the common set of interacting proteins suggested that tsRNAs largely interact with the proteins involved in translation (Fig EV4A). In order to independently validate the previously reported role of tRNA fragments in translation repression [6,15,16], we used an uncapped luciferase mRNA or a capped GFP mRNA in an *in vitro* biochemically defined, rabbit reticulocyte lysate system to measure the translation activity in the presence of exogenously added 5′-tsRNAs. Indeed, we observed an inhibition of protein translation upon the addition of a tsRNA mimic (Fig EV4B and B′). We further validated the interaction of 5′-tsRNA with the translating units by sequencing fractions obtained from polysome profiling in mESCs grown under stem or differentiating conditions (Fig EV4C and C′). Surprisingly, we observed the association of tsRNA with individual subunits as well as the monosome and polysome complexes, suggesting that 5′-tsRNAs may regulate various stages of translation.

To study the differential interacting partners of tsRNAs between LIF- vs. RA-treated mESCs, we chose tsGlnCTG as the representative tsRNA. We performed mass spectrometry to identify tsGlnCTG-associated proteins from LIF- v/s RA-treated mESCs along with a scrambled sequence as a negative control (Fig EV4D). Proteomic analysis revealed that tsGlnCTG interacted differentially with several ribosomal proteins and RNA-binding proteins (RBPs) in stem vs. differentiating state (Fig 4B and Dataset EV4). We validated our proteomic analysis by performing Western blot analysis of candidate proteins such as IGF2BP1, YBX1, and RPL10 that interact with tsGlnCTG (Fig EV4F). Both IGF2BP1 and RPL10 showed similar enrichment as seen in the proteome analysis.

Given that tsGlnCTG differentially interacts with proteins in LIF- and RA-treated conditions, we next investigated the differential interaction of the tsRNA with putative mRNA transcripts between these conditions. Transcriptome analysis of tsGlnCTG pulldown revealed distinct association of transcripts (*P* value < 0.05) with the tsRNA in stem (LIF) vs. differentiated (RA) cell states (Fig 4C). We identified 582 transcripts that associated equally with tsGlnCTG in both stem and differentiated states (Dataset EV5). GO annotations of

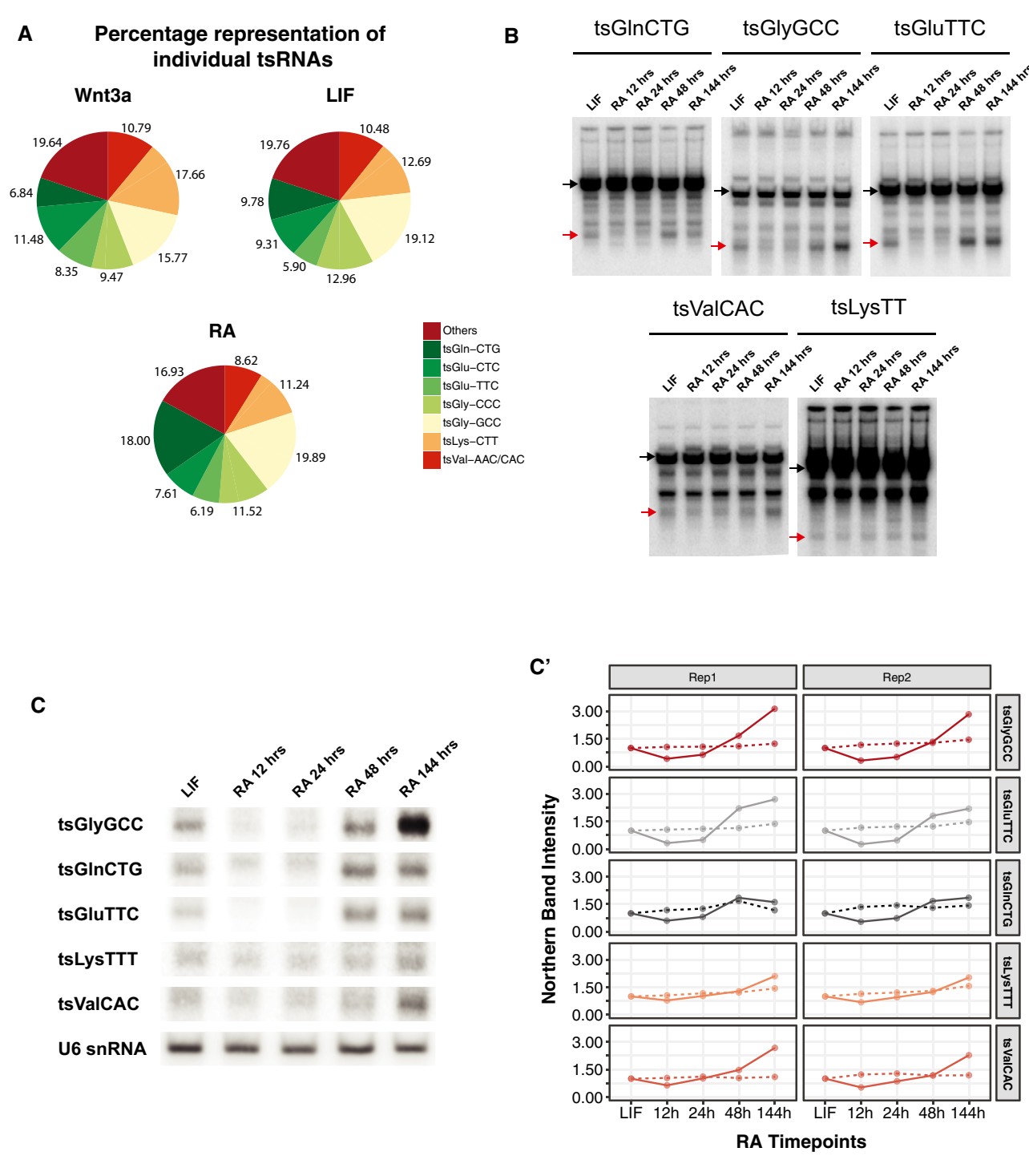

**Figure 2. Characterization of 5-tsRNAs during mESC differentiation.**

A   Pie chart depicting the various species of 5′-tsRNAs expressed in the three different states of stem cells tested.

B   Northern blot of candidate tRNAs and tsRNAs tsGlnCTG, tsGlyGCC, tsGluTTC, tsValCAC, and tsLysTTT. The black arrows represent the tRNAs, and the red arrows mark the tsRNAs.

C   Northern blot of 5′-tsRNAs: tsGlyGCC, tsGlnCTG, tsGluTTC, tsLysTTT, and tsValCAC at various time points of RA-induced mESCs differentiation showing dynamic expression of these 5′-tsRNAs.

C′  Relative quantification of the tsRNA bands from northern hybridization tested across two biological replicates. The solid lines represent the expression of tsRNAs, and the dashed lines represent tRNAs.

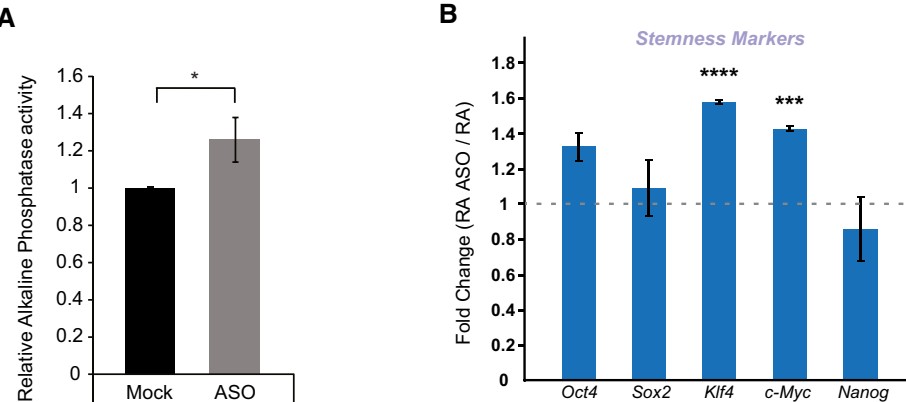

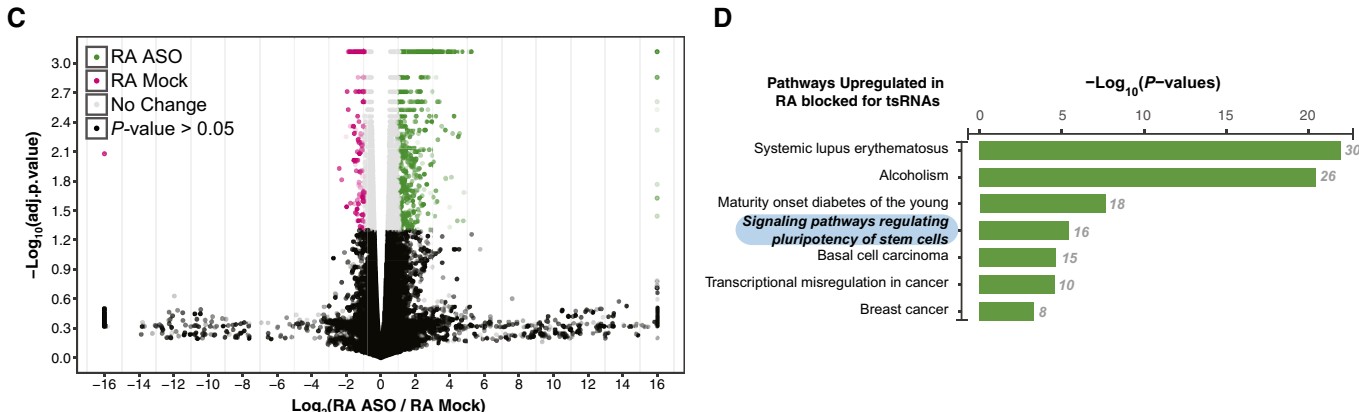

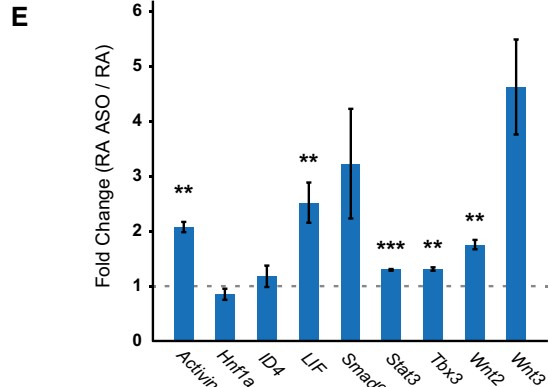

**Figure 3. Effect of 5′-tsRNA knockdown on RA-induced differentiation of mESCs.**

A  Effect of ASO-mediated inhibition of 5′-tsRNAs on alkaline phosphatase activity in RA-treated mESCs. (*n* = 3), error bars represent SEM, and significance was calculated using unpaired *t*-test.

B  Relative expression of stemness markers in RA-induced differentiation mESCs blocked for 5′-tsRNA function with ASOs. (*n* = 2), error bars represent SEM, and significance was calculated using unpaired *t*-test.

C  Volcano plot of the transcriptome depicting the overexpressed transcripts (in green) and downregulated transcripts (in pink) in ASO-treated differentiating cells.

D  KEGG pathway analysis of the transcripts upregulated in a RA-mediated differentiating mESCs blocked for tsRNA function.

E  qPCR validation of transcripts involved in the stem cell signaling pathway that are upregulated upon 5′-tsRNA knockdowns using ASOs. (*n* = 2), error bars represent SEM, and significance was calculated using unpaired *t*-test.

Data information: \*P value < 0.05; \*\*P value < 0.01; \*\*\*P value < 0.001; \*\*\*\*P value < 0.0001; n.s., not significant.

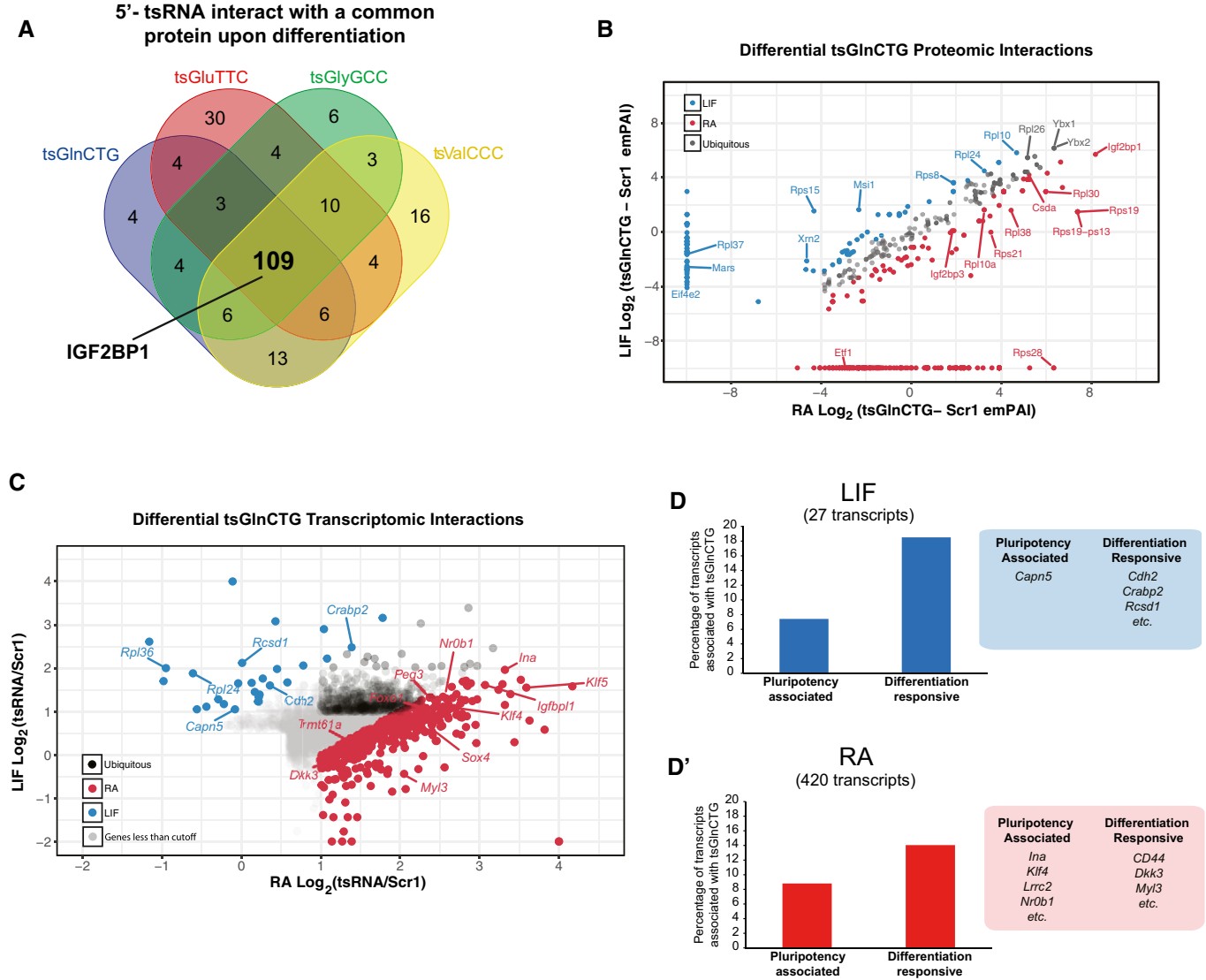

**Figure 4. Identification and functional characterization of tsRNA interactome in mESCs cultured with LIF or RA.**

A  Venn diagram depicting the common interacting proteins between different tsRNAs in RA-treated cells.

B  Scatter plot showing Log₂ fold-change enrichment of protein interactome between LIF- and RA-treated mESCs. All peptides identified by LC-MS/MS had < 1% FDR (Fig EV4E). LC-MS/MS was conducted twice ($R^2$ = 0.85 between LIF duplicates and $R^2$ = 0.82 between RA-treated mESC duplicates).

C  Scatter plot showing Log₂ fold-change enrichment of mRNAs associated with tsGlnCTG over Scramble (Scr1) in LIF- vs. RA-treated mESCs.

D, D′  Bar plots and representative gene lists depicting association of tsGlnCTG with "pluripotency-associated" (D) and "differentiation-responsive" (D′) genes in LIF- or RA-treated mESCs.

these transcripts revealed enrichments of genes involved in many basic cellular processes such as metabolism, and stress response (Dataset EV6). In addition, tsGlnCTG showed enriched associations with 420 transcripts in RA condition vs. 27 transcripts in the stem state, underscoring a potential function for tsGlnCTG in the regulation and/or maintenance of differentiated states (Dataset EV5). To understand the regulatory effect of 5′-tsRNAs on the target mRNAs, we investigated the binding preference of tsGlnCTG toward either "differentiation-responsive" or "pluripotency-associated" genes in both stem and differentiating conditions. Of the 27 tsGlnCTG-associated transcripts that showed enrichment in the stem state, a preference was observed toward "differentiation-responsive transcripts"

(five transcripts) such as *Cdh2*, compared with "pluripotency-associated transcripts" (two transcripts; Fig 4D and Dataset EV5). However, upon RA-mediated mESC differentiation, there was a marked increase (from 2 to 37) in the number of "pluripotency-associated" transcripts associated with tsGlnCTG compared to the stem state (Fig 4D′). Some of these transcripts are *Klf4*, *Ina*, *Nr0b1*, and *Lrrc2*, which are known pluripotency factors critical for the maintenance of the stem state [23]. Interestingly, we also observed an increase in tsGlnCTG association with pro-differentiation transcripts such as *Myl3*, *CD44*, and *Dkk3* [24–26] (Fig 4D′ and Dataset EV5). While initially counter intuitive, upon closer examination of these transcripts, we noticed a significant association (*P* value < 0.05)

with genes involved in skeletal, cardiovascular, and blood vessel development compared with neuronal lineage (Dataset EV6). The fact that pro-neuronal genes were not enriched by 5′-tsRNAs compared with other lineages suggests the intriguing possibility that during neural lineage differentiation triggered by RA [27], tsGlnCTG may gain an additional function in targeting and regulating transcripts driving non-neuronal pathways. Interestingly, when we compared the transcripts that associated with 5′-tsRNAs in the RA condition with transcripts that were either up- or downregulated in RA-treated cells transfected with ASOs, we observed minimal overlap (Fig EV4G). This suggested that the 5′-tsRNAs mainly function through translation regulation of transcripts rather than influencing target stability. Taken together, these data suggest that upon RA treatment tsGlnCTG contributes to stem cell differentiation by influencing distinct subsets of mRNAs important for pluripotency and non-neuronal lineage specification.

The association of 5′-tsRNAs to mRNAs could be either direct (through sequence complementarity) or indirect (through their association with certain RBPs). To verify whether the transcripts enriched by 5′-tsRNAs in our study are protein-dependent or protein-independent interactions, we performed tsGlnCTG pull-downs in normal cell lysates vs. lysates devoid of proteins (Fig EV4H). qPCR analysis revealed that 5′-tsRNAs could potentially interact with transcripts via intermediate proteins as well as through direct RNA–RNA interactions (Fig EV4I). Although bioinformatics analysis failed to identify any significant miRNA-like complementarity between tsGlnCTG and the target mRNAs, it does not preclude the possibility that other 5′-tsRNAs may potentially interact with mRNAs with unique complementarity signatures.

## 5′-tsRNAs regulate c-Myc transcript levels through association with Igf2bp1

As shown in Fig 4, in RA-treated mESCs, we observed a strong association of tsGlnCTG with the RBP, Igf2bp1/Imp1 (Figs 4A and B, and EV4J, and Dataset EV4), an important protein required for stem cell maintenance and function [28,29]. Although the levels of Igf2bp1 over 48 h of RA-induced differentiation remain constant (Fig EV5A), RNAi-mediated knockdown of Igf2bp1 led to a loss of pluripotency (Fig EV5B and B′). Interestingly, Igf2bp1 is known to prevent endonucleolytic cleavage of c-Myc mRNA by binding to the coding region instability determinant (CRD), thus facilitating efficient c-Myc mRNA association with the translating pool [30]. c-Myc

is a regulator of cell states; its overexpression induces pluripotency [31], and its loss results in entry into a diapause state [32]. We hypothesized that the change in interactions of Igf2bp1 with c-Myc could be crucial for maintaining pluripotency and/or differentiation. Therefore, we investigated whether the increased association of 5′-tsRNAs with Igf2bp1 perturbs its binding to c-Myc mRNA, thereby facilitating the transition of mESCs from a pluripotent to a differentiated state.

We performed RNA immunoprecipitation (RIP) assays to study the interaction of Igf2bp1 with 5′-tsRNAs and the c-Myc transcript. UV-RIP followed by small RNA sequencing from RA-treated mESCs revealed associations of Igf2bp1 with a large proportion of tsRNA (Fig EV5C) including tsGlnCTG (Dataset EV7 and Fig EV5D). We further validated this interaction by performing *in vitro* binding assays. We observed that 5′-tsRNAs directly interact with the Igf2bp1 protein with a $K_d$ of 33 nM (Figs 5A and EV5E). Additionally, by performing real-time PCR analysis of the Igf2bp1 RIP, we observed a decreased association of Igf2bp1 with *c-Myc* mRNA under differentiation conditions compared with LIF (Figs 5B and EV5F). To investigate the function of 5′-tsRNAs in the regulation of *c-Myc*-Igf2bp1 association, we blocked 5′-tsRNAs using ASOs during RA-induced differentiation. Strikingly, the addition of ASOs disrupted the interaction of tsRNA with Igf2bp1 as evident from the *in vitro* binding assay (Fig 5A). Notably, polysome profiling revealed decreased levels of *c-Myc* mRNA association with actively translating pools (polysomes and 80S) in RA-treated cells compared with LIF-treated mESCs (Fig 5C). However, when 5′-tsRNAs were blocked using ASOs in RA-treated mESCs, we observed a marked increase in *c-Myc* association with translating pools with a simultaneous decrease in the non-translating pool (Fig 5D). This was further validated by the observation that the overall levels/stability of *c-Myc* mRNA increased in RA-ASO compared with RA-treated conditions (Figs 5E and 3B). To further elucidate the antagonistic role of 5′-tsRNAs in blocking the interaction between Igf2bp1 and *c-Myc*, we used HCT116 Myc-Luc reporter cell lines to measure the luciferase activity driven by a c-Myc-regulated promoter. The extent of luciferase activity is a direct measure of levels of active c-Myc protein. We transfected tsGlnCTG into the HCT116-Myc-Luc reporter cell line in the background of IGF2BP1 siRNA knockdown. The transfection of tsGlnCTG mimic was sufficient to inhibit luciferase activity by ∼ 30%, which was further reduced by an additional 10% upon IGF2BP1 knockdown (Fig 5F). Taken together, our data suggest a mechanism for tsRNA function that involves

**Figure 5.  5′-tsRNA-mediated regulation of c-Myc through Igf2bp1 interaction..**

A  *In vitro* binding analysis of tsGlnCTG to Igf2bp1 in the presence and absence of antisense oligo (ASO) against tsGlnCTG. ASOs effectively disrupt the binding of tsGlnCTG to Igf2bp1. (*n* = 2) error bars represent SEM.

B  Quantification of the Igf2bp1-bound *c-Myc* mRNA between LIF- vs. RA-treated mESCs (*n* = 2; see Fig EV5F for duplicate data).

C  Association of c-Myc transcript in different translating pools in RA-treated mESCs as compared to LIF condition. (*n* = 2), error bars represent SD, and significance was calculated by one-tailed unpaired *t*-test.

D  Relative enrichment of *c-Myc* mRNA in translating (80S and polysome) and non-translating (mRNPs) pools (fractionated from polysome profiling) between ASO-treated and mock-treated RA-induced differentiating mESCs. (*n* = 2), error bars represent SD, and significance was calculated using one-tailed unpaired *t*-test.

E  Relative levels of *c-Myc* mRNA in ASO-treated and mock-treated RA-induced differentiating mESCs compared with LIF-treated mESCs. (*n* = 3), error bars represent SD, and significance was calculated using one-tailed unpaired *t*-test.

F  Epistatic analysis of 5′-tsRNAs and IGF2BP1 in regulating Myc transcriptional reporter activity (*n* = 3), error bars represent SD, and significance was calculated using unpaired *t*-test.

G  Schematic representing tsRNA based c-Myc transcript regulation.

Data information: *P value < 0.05, **P value < 0.01; ***P value < 0.001.

sequestering of Igf2bp1 from the c-Myc mRNA–protein complex. This in turn results in decreased expression and translation of c-Myc mRNA, thereby driving and/or facilitating the maintenance of the differentiated state in RA-treated mESCs (Fig 5G).

## Discussion

Transition and maintenance of defined cell states are fundamental to normal development. The process of stem cell differentiation into

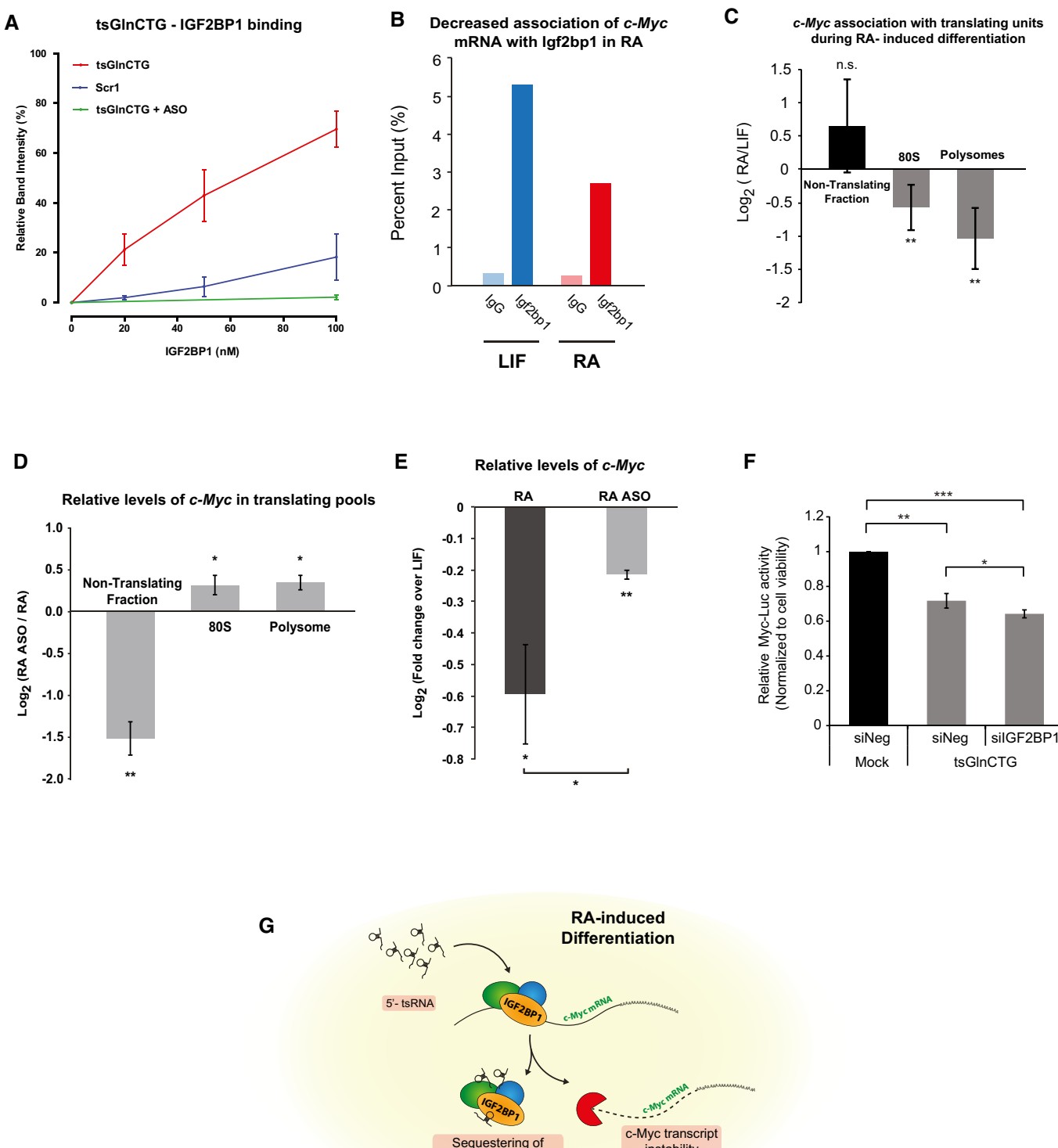

**Figure 5.**

lineage-specific progenitors involves changes at both transcriptional and translational levels. In this present study, we probed tsRNA expression in a variety of heterologous models of stem vs. differentiated states, and observed a preferential enrichment of tsRNAs in the more differentiated cell types. We investigated the nature of these tsRNAs across the context of various cell-state transitions and observed that the majority of tsRNAs specifically and consistently matched to the 5′-half of tRNAs. Although several studies have identified diverse roles for tsRNA species in various biological contexts [7,9,10,12], our study specifically identifies a role for 5′-tsRNAs in modulating RA-mediated differentiation of mESCs by regulating translation and the stability of target mRNAs.

Intriguingly, our bioinformatic analysis suggests that only a small subset of the 468 tRNA loci contribute to the pool of 5′-tsRNAs. This evokes the requirement of specific processing of certain tRNAs raising the question of how that might be achieved. An important protein that is thought to be involved in the processing of 5′-tsRNAs is angiogenin. Several reports have shown the involvement of angiogenin in cleaving tRNAs during stress response resulting in a 3′ cyclic phosphate [5,6,14,33,34]. It is interesting to note that we observed a large population of tsRNAs through conventional sequencing strategies, which cannot capture sequences with 3′ cyclic phosphate ends. This suggests an angiogenin-independent mechanism for the processing of tRNAs. Additionally, sequencing of T4 PNK-treated RNA isolated from RA-induced differentiated mESCs yielded only a modest increase in the 5′-tsRNA population, further suggesting a minor role for angiogenin in the processing of 5′-tsRNAs characterized in our study. A comprehensive functional screen would be necessary to identify endonucleases essential for the processing of 5′-tsRNAs.

It has been established that 5′-tsRNAs are involved in translational repression [15,16]. In our study, we also observed a significant enrichment of 5′-tsRNAs in monosome and polysome fractions during differentiation, suggesting that 5′-tsRNAs may regulate translation at multiple steps in addition to their reported role in repressing translational initiation [15]. In concordance with these reports, our *in vitro* translation studies support the role of tsRNAs as general translational repressors. However, our *in vivo* biochemical assays revealed specificity in their interactions with target transcripts. Notably, tsRNAs associated with "differentiation-responsive" transcripts in the stem state, suggesting their function in translational repression of transcripts that are critical for maintenance of the stem state. Likewise, during RA-induced differentiation, 5′-tsRNAs interact with both "pluripotency-associated" and "differentiation-responsive" transcripts. Although the association of 5′-tsRNAs with "differentiation-responsive" transcripts is counter intuitive, GO annotation revealed that the transcripts were enriched for several lineages apart from those promoting or associated with neuroectoderm specification. Taken together, our data suggest an additional role of 5′-tsRNAs in the specific regulation of transcripts that are critical in stem cell maintenance and for lineage specification during cellular differentiation (Fig 6).

An interesting question raised by our study is how modest changes in levels of tsRNA expression elicit biological response. While we found only a 15% increase in the overall levels of tsRNAs (in RA-treated condition compared with LIF), our data revealed increased association of tsRNAs with transcripts in RA condition compared with LIF (27 vs. 420). This suggests that there could be additional factors that facilitate specific interactions of tsRNAs with their targets in RA conditions. Notably, our study reveals two possible modalities for target selection, either via direct interactions with target transcripts or through its interaction with specific set RBPs and ribosomal proteins. Our pulldown data from lysates devoid of proteins revealed that tsRNAs can also directly interact with target mRNAs, perhaps through sequence complementarity. However, bioinformatic analyses failed to identify any canonical miRNA-like

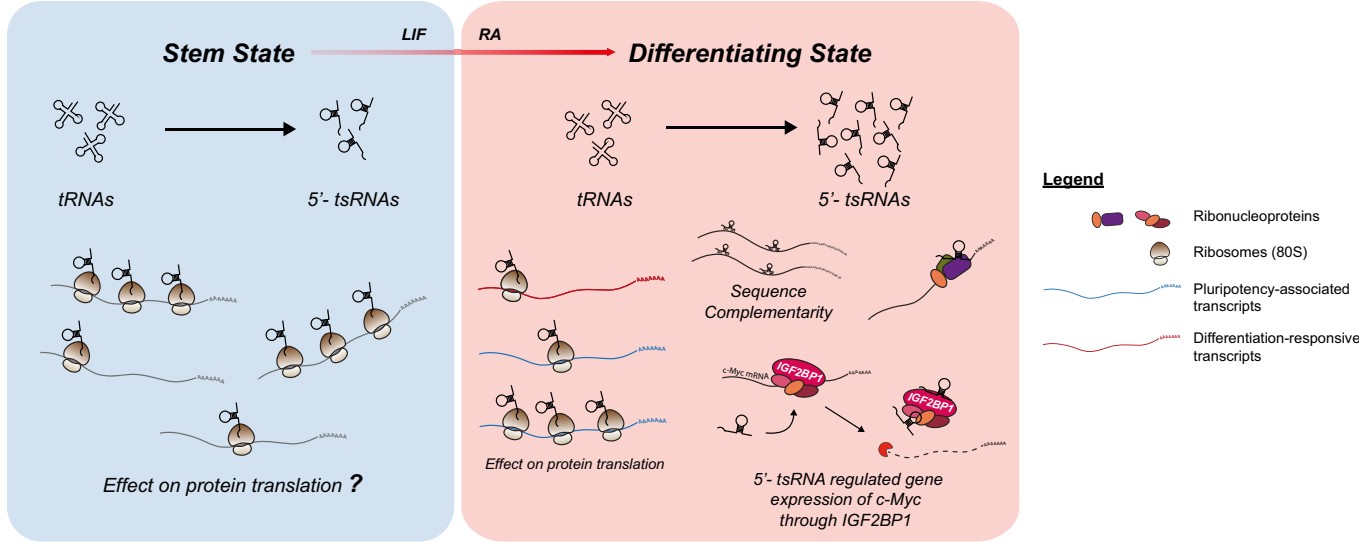

**Figure 6.  Proposed mechanistic model for tsRNA cellular and molecular functions.**

The dynamic expression of 5′-tsRNAs plays a crucial role in modulating stem cell differentiation. During differentiation, 5′-tsRNAs regulate the translation and/or the stability of several transcripts through its interaction with ribosomes, RNA-binding proteins (ribonucleoproteins), such as IGF2BP1, or through direct sequence complementarity.

"seed" region (2–8 nt) in the tsRNAs. Furthermore, we failed to observe any association of tsRNAs with proteins involved in the miRNA pathway in the mass spectrometry-based pulldown assay, suggesting the existence of an as yet unidentified target recognition module exhibited by tsRNAs. Additionally, the association of 5′-tsRNAs with different sets of ribosomal proteins between stem and differentiating cells invokes two exciting possibilities for tsRNA function. First, 5′-tsRNAs may confer functional heterogeneity to ribosomes by associating with specific pools of ribosomal proteins that are involved in the translation of specific targets. Second, the spatial positioning of 5′-tsRNAs on the ribosome, evident from its interaction with both large and small subunits, suggests different modes of translational regulation. In a recent study conducted in *Halofex volcani*, a 26 nt 5′-tRF derived from tRNA$^{Val}$ was shown to bind to a region of small subunit of the 70S ribosome, thereby competing with the mRNA leading to inefficient translation [35]. While the ~ 30 nt tsRNAs identified in our study could potentially function through similar mechanisms, detailed structural studies will be required to better understand the functional relevance of tsRNA associations with ribosomal proteins.

In this study, we focused on importance of the interaction between tsRNAs and Igf2bp1, and its functional consequences on the regulation of c-Myc, a well-known pluripotency factor. Igf2bp1 is an RBP implicated in stem cell maintenance and carcinogenesis that regulates the stability and translation of several mRNAs, including c-Myc [28,30,36]. The tsGlnCTG–protein interactome revealed enhanced binding to Igf2bp1 in RA conditions leading us to hypothesize that tsGlnCTG could potentially regulate *c-Myc* through its association with Igf2bp1. Indeed, we found decreased association of the c-Myc transcript with Igf2bp1 in RA-induced differentiating cells, suggesting that the increased interaction of 5′-tsRNAs with Igf2bp1 could potentially destabilize the Igf2bp1-*c-Myc* complex. Loss-of-function studies using ASOs targeting tsGlnCTG resulted in increased c-Myc mRNA stability and its association with the translating pool. This further validated our hypothesis that tsGlnCTG, and perhaps other Igf2bp1-binding 5′-tsRNAs, can dictate the fate of c-Myc mRNA via their interaction with Igf2bp1. This was further supported by gain-of-function studies using tsRNA mimics, where we observed an inhibition of c-Myc-driven luciferase reporter activity. However, it is important to note that the effects of tsRNAs are subtle. We believe that these subtle changes are sufficient to facilitate and/or maintain changes in stem states. For example, in mouse ESCs, it has been shown that moderate alterations of Oct4 levels could lead to reduced pluripotency resulting in differentiation [37]. Similarly, small perturbations in c-Myc levels can alter the proliferative status of cells [38]. Such examples emphasize the need for a regulatory program that can sustain critical levels of proteins essential for the maintenance of cell states. We believe that tsRNAs may be involved in fine-tuning translation and target stability of cell-state-associated transcripts, thereby altering their protein expression levels. Altogether, our results suggest that 5′-tsRNAs, including tsGlnCTG, may play critical functions in regulating c-Myc mRNA stability through their interactions with RBPs such as Igf2bp1 to modulate stem cell differentiation (Fig 6). Furthermore, in concordance with previously reported work, the 5′-tsRNAs in our model system also interact with other RBPs including Ybx1, suggesting that tsRNA-YBX1 interaction may also be involved in regulating transcripts essential for both self-renewal and differentiation. Therefore,

additional probing of tsRNA interactions with other RBPs will likely provide a more comprehensive understanding of tsRNA function during stem cell differentiation.

Taken together, our data highlight the role of 5′-tsRNAs in regulating translation critical for defining cell states. The enrichment observed in 5′-tsRNA levels in differentiated states may also serve as potential biomarkers for identifying cell states during differentiation and in cancer progression. Given that tsRNAs facilitate state transitions, understanding the underlying mechanisms, specifically in cancer models, could aid in developing future therapies to counter cancer progression.

# Materials and Methods

### Cell culture

E14 mouse embryonic stem cells (mESCs) were cultured in high glucose DMEM media, supplemented with 15% ES grade fetal bovine serum (FBS; Biowest S182S-500, Batch S10397S182), sodium pyruvate, gentamycin (Gibco 15710-064), non-essential amino acids (NEAA; Gibco 11140-050), glutamax (Gibco 35050-061), β-mercaptoethanol (Gibco 21985023), and 1,000 U/ml LIF (ESGRO 1106). The mESCs were grown on 0.1% gelatin-coated plastic and passaged every 2 days, with daily media changes, and for a maximum of 12 passages. mESCs were differentiated with 0.5 μM RA for 48 h, unless specified otherwise. HCT116 Myc-Luc reporter cells were purchased from Amsbio, BPS Bioscience (#60520). MDA-MB-231 and HS578T breast cancer cell lines were gifts from Wai Leong TAM. The above were cultured in high glucose DMEM with sodium pyruvate, supplemented with 10% FBS (Biowest S181B-500), 1% penicillin/streptomycin (Gibco 15140-122), and 1% glutamax (Gibco 35050-061). The HCT116 Myc-Luc reporter line is additionally cultured under selection with 500 μg/ml geneticin. Human mammary epithelial cells (HMECs) 1–3 were a gift from Mathijs Voorhoeve. The HMECs were maintained in MEGM™ Mammary Epithelial Cell Growth Medium (Lonza MEGM BulletKit; CC-3151 and CC-4136).

### RNA isolation

Total RNA from cells, including small RNAs, was isolated through phase separation with TRIzol according to manufacturer's recommendations, followed by capture with the miRNeasy kit (Qiagen 217004).

### Quantitative real-time PCR (qRT–PCR)

Total cellular RNA was extracted as described above. A minimum of two biological replicates were used per sample. Reverse transcription (RT) was conducted with the SuperScript II Kit (Invitrogen) according to the manufacturer's protocol. qPCR utilizing the KAPA SYBR FAST qPCR mix (KAPA Biosystems) was conducted with the QuantStudio® 7 Flex Real-Time PCR System (Thermo Fisher Scientific). Primers for qRT–PCR are listed in Dataset EV8.

### Mouse skin isolation and FACS

Back skin of four 8 weeks CD1 mice was trypsinized overnight at 4°C. The epidermis was scraped and chopped into fine pieces to

further trypsinize to isolate single cells. The cells were washed with PBS and resuspended in PBS + 1% FBS. Cells were stained for 1 h on ice with integrin α6-PE (ab95703, Abcam) and/or CD34-FITC-conjugated (11-0341-85, eBioscience) antibodies. Cells were sorted on FACS Aria™ (BD Biosciences) based on CD34$^+$α6$^+$ (bulge stem cells), CD34$^-$α6$^+$ (basal epithelia), and CD34$^-$α6$^-$ (differentiated keratinocytes; Appendix Fig S1). This was followed by RNA extraction and small RNA library prep (TruSeq Small RNA Prep Kit, Illumina). Small RNA libraries were sequenced on Illumina NextSeq 500 platform.

MDA-MB-231 and HS578T breast cancer cells were sorted for CD44/CD24 expression with Alexa Fluor 488-conjugated α-CD44 (Stem Cell, Clone IM7, Lot #SC02931) and α-CD24 (Santa Cruz Biotechnology, FL-80, SC-11406 Lot #E1413). Cells were sorted with the FACS Aria™ Fusion Sorter (BD Biosciences; Appendix Figs S2 and S3) operated by the A*STAR SIgN common FACS facility. Each sample was analyzed with negative controls and single channel/color controls.

## Microfluidic RNA gel electrophoresis

Equal amounts (not exceeding 100 ng) of total RNA per sample were loaded in each small RNA chip well (Agilent Small RNA kit, 5067-1548), in accordance with product protocols, and analyzed with the Agilent 2100 Bioanalyzer. Nucleotide size regions of interest were gated, and subsequent quantitation of RNA abundance of specified regions was obtained using the Bioanalyzer Expert software.

## Small RNA library preparation

All libraries generated for small RNA sequencing were prepared using TruSeq Small RNA Prep Kit (RS200012, RS200024, RS200036, RS200048, Illumina) using the manufacturer's protocol. 1 μg of RNA was used for the preparation of the libraries. The PCR-amplified libraries were run on an 8% polyacrylamide gel. The bands between 140 and 165 bp were excised, and the cDNA was extracted. The libraries were multiplexed and sequenced on Illumina NextSeq 500 platform. For T4 PNK-treated libraries, the RNA was treated with T4 PNK (without ATP) at 37°C for 40 min. The treated RNA was TRIzol extracted, and 1 μg of RNA was taken for small RNA library preparation.

## Identification and characterization of tsRNAs

We used mm10 [39] and hg19 [40] as the reference genomes (http://genome.ucsc.edu/) for mouse and human samples, respectively. From the sequencing reads, we trimmed TruSeq small RNA adapters using customized perl script and cutadapt program (-f fasta -b TGGAATTCTCGGGTGCCAAGG -O 5 -m 6 -o) [41]. The adapter-trimmed reads were aligned to rRNA, and unaligned reads were taken for further analysis. For analysis, we considered reads ranging between 18 and 35 nucleotides and mapped these to genome and the tRNA database obtained from GtRNAdb [42] using bowtie v1.1.2 (-f -v 2 -p 20 –un) [43]. We used two mismatches as a constant parameter for all the alignments done in this study, as we did not observe much deviation in our results with varying mismatches ranging from 0 to 2. Reads, which are 30–35 nucleotides in length and specifically map only to tRNA database, were considered as tsRNAs. Since multiple loci on the genome code for

the same tRNA isotype, and to avoid multi-mapping problem (same reads mapping to multiple locations), we clustered the tRNA based on sequence similarity of the first 50 nucleotides (468 tRNAs were clustered to 226 unique tRNA sequences). We have provided the cluster information in Dataset EV2. We did not consider 18–24 mer reads as tsRNAs because only negligible reads mapped to tRNA database. To calculate the percentage tsRNA reads that mapped to genome, we calculated the ratio of reads of a particular size that mapped to tRNA to the total reads that mapped to the genome (Fig 1B). Further, we calculated per base tRNA coverage using the following formula: *per base coverage* = (number of reads aligned to particular base of tRNA/total number of reads mapped to all the tRNA in that sample) which is plotted in Fig 1D.

We used VARNAv3-93 [44] for obtaining secondary structures of tRNA, and we used the coverage values obtained to plot the tRNA heat maps. We used the well-established DESeq algorithm [45] for normalization of the sequencing data and identification of differentially expressed 5′-tsRNAs (adj. *P*-value < 0.05). All the statistical tests were done in R [46]. We used customized perl script for all the analysis used in this study. We used R ggplot2 [47] library for plotting. The raw numbers and mapping percentage for all the sequenced samples are given in Dataset EV1.

## Northern hybridization

The RNA blot was performed as described by previously [48]. 10 μg of total RNA was resuspended in 8 μl loading buffer (0.10% bromophenol blue, 0.10% xylene cyanol in 100% de-ionized DEPC-treated formamide), heated at 95°C for 1 min, and loaded on to a 15% denaturing polyacrylamide gel (a 19:1 ratio of acrylamide to bisacrylamide, 8 M urea). The gel was run at 100 V for 3 h and then transferred to a Hybond N+ membrane by electroblotting at 10 V overnight at 4°C. The hybridization was performed at 35°C for 12 h in UltraHyb-Oligo buffer (Ambion). DNA oligos complementary to candidate 5′-tsRNAs (Dataset EV8) were end-labeled with $^{32}$P-ATP (Board of Radiation and Isotope Technology, India) using polynucleotide kinase (NEB), purified through MicroSpin G-25 Columns (GE Healthcare), and were used as probes. The blot was washed twice with 2× SSC, 0.5% SDS for 30 min at 35°C. The signal was detected after exposure on a phosphorimager screen using a Molecular Imager.

## Small RNA qPCR

Total RNAs were isolated from mESCs (LIF- or RA-treated). At least two biological replicates were used per sample. For each sample, 1 μg of RNA was then reverse-transcribed with Superscript II RT (Invitrogen) using tsRNA-specific stem-loop RT primers (Sigma; Dataset EV8). Reverse-transcribed cDNA was amplified using Maxima SYBR Green/ROX qPCR Master Mix (K0222, Fermentas) using tsRNA-specific forward primer and universal reverse primer against the stem-loop RT primer (Dataset EV8). Sno-202 was used as an endogenous control to normalize differences. The samples were amplified and analyzed on Applied Biosystems 7900HT qPCR machine.

## RNA oligonucleotides

All RNA oligonucleotides were synthesized and purchased from Integrated DNA Technologies. RNA oligonucleotides against

sequences of candidate 5′-tsRNAs are 2′OMe modified. Chimeric antisense oligonucleotides (ASOs) are complementary phosphorothioate-modified DNA sequences flanked by 2′OMe modified RNA. Pool of five ASOs against tsGlyGCC, tsGlyCCC, tsGlnCTG, tsValCAC, and tsArgTCT was used to investigate the effects of tsRNA knockdown. 3′-Biotin with TEG linkers were tagged to tsGlnCTG or a nonsense-scrambled control (Scr1). Sequences of oligonucleotides used are found in Dataset EV8.

### Transfections of mESCs

$3 \times 10^5$ or $6 \times 10^5$ mESCs per well (6-well plate) were plated for forward and reverse transfections, respectively. mESCs were transfected with Lipofectamine 2000 (Thermo Fisher Scientific) according to manufacturers' recommendations. mESC transfection media consist of 1:1 parts of 2× mESC media supplemented with LIF, and Opti-MEM Reduced Serum Media (Gibco 31985070). Transfection media were changed to mESC media without antibiotics 5–6 h or 16–18 h post-transfections for forward and reverse transfections, respectively.

Alternatively, for large-scale experiments such as polysome profiling followed by qPCR, cells were transfected in Neon Transfection System (Thermo Fisher, MPK10096) at 1,100 V for three pulses. 100 nM concentrations of each tsRNA antisense oligo (ASO) were used for transfection.

### Alkaline phosphatase activity assay

Alkaline phosphatase activity of mESCs was measured using the Alkaline Phosphatase Assay Kit (Colorimetric; Abcam ab83369), in accordance with product-recommended protocol. The mESCs were grown and transfected in 48-well TC plates. Optical densities at 405 nm were read with the Tecan Infinite M1000 PRO Multimode Microplate Reader in 96-well clear flat bottom TC plates.

### Analysis of transcriptome (LIF overexpressed with 5′-tsRNA or 5′-tsRNAs blocked with ASOs in RA condition)

We performed single-end RNA sequencing (1 × 100 bp) for mESC treated with tsRNA (under LIF condition) or ASOs (under RA condition) along with mock controls. We obtained ~ 41–49 million reads per sample. Adapters were trimmed using Trimmomatic [49] and subsequently mapped to rRNA. Reads that did not align to rRNA were taken for further analysis. We used reference-based transcriptome assembly algorithms Hisat2 v2.1.0 [50]; Cufflinks v2.2.1 [51]; and Cuffdiff v2.2.1 [52] to identify differentially expressed transcripts. We aligned the reads to mouse (mm10) genome using Hisat2 (-q -p 8 –min-intronlen 50 –max-intronlen 250000 –dtacufflinks –new-summary –summary-file). Around 93–95% of reads mapped to reference genome. The mapped reads were assembled using Cufflinks using mm10 Refseq gtf file. Transcripts which had adjusted *P*-values < 0.05 and minimum twofold up/downregulated were considered as significantly expressed and taken for further analysis. We identified 3,632 transcripts with significant pvalue between RA treated with mock and ASO from cuffdiff, among which 1,043 transcripts were twofold up/downregulated. We also identified 189 transcripts between LIF treated with mock and tsRNA from cuffdiff, among which 69 transcripts were twofold up/

downregulated with significant *P*-value. We performed pathway analysis and gene-ontology analysis for these selected up/downregulated transcripts using g-profiler [53]. We used customized perl script for all the analysis used in this study. We used R ggplot2 [47] and CummeRbund [54] library for plotting.

### Biotinylated RNA mimics pulldown

Pulldowns with 3′-biotinylated RNA mimics were conducted with a protocol adapted from Ref. [55]. 3 μg of folded biotinylated tsGlnCTG or a scrambled sequence (Scr1), in 100 μl of RNA structure buffer (10 mM Tris pH 7.0, 0.1 M KCl, 10 mM MgCl$_2$), was incubated with 1 mg of pre-cleared cell extracts in cell lysis buffer (20 mM Tris–HCl pH 7.5, 150 mM NaCl, 1.8 mM MgCl$_2$, 0.5% NP40, protease inhibitor cocktail for mammalian cells (no EDTA), 1 mM DTT, 80 U/ml RNase inhibitor). Proteins were eluted with SDS buffer from streptavidin agarose beads and analyzed on PAGE. RNA was extracted from Dynabeads M-280 Streptavidin™ (11206D, Invitrogen) beads using TRIzol (Invitrogen). Isolated RNA was analyzed on microfluidic RNA gel electrophoresis with the Agilent 2100 Bioanalyzer before RNA sequencing.

For analyzing interactions of tsRNA with transcripts, an equal number of cells were taken for lysis (using 1 ml lysis buffer—whole cell lysates) and TRIzol RNA extraction. The extracted RNA was resuspended in 1 ml of lysis buffer, RNA lysate (devoid of protein). Pulldowns of tsGlnCTG were carried out from cell lysates as described above. The RNA precipitated by tsRNA was dissolved in 10 μl water, of which 1 μl was taken for reverse transcription. The $C_t$ values obtained from RNA-lysate pulldowns were compared with $C_t$'s obtained from whole cell lysates. For validating protein interaction through westerns, 1× Laemmli buffer was added to the pulldown samples and boiled at 95°C for 10 min and loaded on to a SDS–PAGE. This was followed by a transfer onto a nitrocellulose membrane and subsequently probed with antibodies against Igf2bp1 (8482, CST), Ybx1 (ab12148, Abcam), or Rpl10 (ab110371, Abcam).

### Protein elution and in-gel tryptic digestion

The beads used to pull down the RNA–protein complex were boiled in gel loading buffers. The eluted samples were run on a SDS–PAGE gel. Gel slices were cut into small pieces and transferred to Eppendorf tubes. They were washed for several times with Milli-Q water, and then followed with 50% ACN/50% 25 mM NH$_4$HCO$_3$ via vigorous vortexing for 30 min. The gel pieces were then dehydrated with 100% ACN. They were then reduced with 10 mM DTT (dithiothreitol) at 56°C for 1 h and alkylated with 55 mM IAA (iodoacetamide) for 45 min in the dark followed by successive washes with 25 mM NH$_4$HCO$_3$ and 50% ACN/50% 25 mM NH$_4$HCO$_3$ with vigorous vortexing for 30 min. The gel pieces were dehydrated again with 100% ACN. Trypsin (V5111; Promega, Madison, WI) was added in the ratio of 1:50. After the trypsin solution was completely absorbed by gel particles, 25 mM NH$_4$HCO$_3$ was added to completely cover the particles. They were then incubated at 37°C overnight. Peptides were extracted from gel particles with 50% ACN containing 0.1% TFA under sonication for 30 min twice. The combined extracts were dried in vacuum and stored at −20°C before LC-MS/MS analysis.

## LC-MS/MS

LC-MS/MS was performed as described previously [56]. Tryptic peptides were dissolved in 0.1% formic acid (FA) in 2% acetonitrile (ACN). They were then analyzed on a Dionex Ultimate 3000 RSLCnano System coupled to a Q Exactive tandem mass spectrometer (Thermo Fisher, CA). Each peptide sample was injected into an acclaim peptide trap column (Thermo Fisher, CA) via the auto-sampler of the Dionex RSLCnano System. Peptides eluted from the peptide trap were separated in a Dionex EASY-spray column (PepMap® C18, 3 μm, 100A, 75 μm × 15 cm; Thermo Fisher, CA) at 35°C. Mobile phase A (0.1% FA in $H_2O$) and mobile phase B (0.1% FA in 100% ACN) were used to establish a 60-min gradient at a flow rate of 300 nl/min. Peptides were then analyzed on Q Exactive with an EASY nanospray source (Thermo Fisher, MA) at an electrospray potential of 1.5 kV. A full MS scan (350–1,600 m/z range) was acquired at a resolution of 70,000 at m/z 200 and a maximum ion accumulation time of 100 ms. Dynamic exclusion was set as 30 s. Resolution for HCD spectra was set to 35,000 at m/z 200. The AGC setting of full MS scan and $MS^2$ were set as 1E6 and 2E5, respectively. The 10 most intense ions above a 1,000 counts threshold were selected for HCD fragmentation with a maximum ion accumulation time of 120 ms. Isolation width of 2 Th was used for $MS^2$. Single and unassigned charged ions were excluded from LC-MS/MS. For HCD, normalized collision energy was set to 28%. The underfill ratio was defined as 0.1%.

## LC-MS/MS data analysis

The raw data were processed as previously described [57]. Briefly, raw data files were converted to the mascot generic file format using Proteome Discoverer version 1.4 (Thermo Electron, Bremen, Germany) with the MS2 spectrum processor for de-isotoping the LC-MS/MS spectra. The concatenated target-decoy UniProt [58] human database (sequence 88 473, downloaded on November 29, 2013) was used for LC-MS/MS spectra searches. The database search was performed using an in-house Mascot server (version 2.4.1; Matrix Science, Boston, MA) with MS tolerance of 10 ppm and LC-MS/MS tolerance of 0.02 Da. Two missed trypsin cleavage sites per peptide were tolerated. Carbamidomethylation (C) was set as a fixed modification, while oxidation (M) and deamidation (N and Q) were variable modifications. Label-free quantitation of proteins was based on emPAI values of each identified proteins reported by Mascot. The relative protein quantities between samples were then calculated from the emPAI values of the protein in the samples.

## Analysis of tsRNA proteomic associations

Non-detectable emPAI (protein abundance index) values (−1) were replaced with lowest detected values (0.01). Beads emPAI values were then subtracted from tsGlnCTG emPAI values. Candidates that display higher emPAI scores in bead controls vs. respective conditions were disregarded. Specific functional enrichment of tsGlnCTG-associated proteins showing more than a twofold change in emPAI scores was investigated for enrichment of molecular signature gene sets (MSigDB, v5.1) [59] using Fisher's exact test, followed by multiple testing correction using false discovery rate (FDR) estimation

[60]. Significant gene sets were determined at FDR-adjusted *P*-value < 0.05 (Dataset EV4).

## *In vitro* translation assay

300 ng of uncapped *Firefly* luciferase mRNA (L4561, Promega) was used in a 25 μl rabbit reticulocyte lysate system (L4960, Promega) and incubated at 30°C for 90 min. Scramble (Scr2) or candidate tsRNA (Dataset EV8) was added to the lysate at a concentration of 0.5 pmole/μl. 1 μl of this reaction was used in a 20 μl Luciferase Assay System (E1500, Promega) to measure relative luciferase counts. Alternatively, EGFP was amplified from pEGFPN3 construct using a T7-tagged forward primer and a reverse primer (Dataset EV8). Capped-polyadenylated GFP mRNA was synthesized using mMESSAGE mMACHINE T7 Ultra Kit (AM1345, Ambion) following manufacturer's protocol. 100–200 ng of the capped-polyadenylated GFP mRNA was used in Retic Lysate System (AM1200, Ambion) and incubated at 30°C for 90 min. Scramble (Scr2) or tsGlnCTG (Dataset EV8) was added to the lysate at varying concentrations. Emission of GFP at 507 nm was measured using a spectrofluorometer (FluoroLog-3, Horiba Scientific).

## Polysome profiling

Mouse ESCs ($4.86 × 10^6$) were seeded in each 15-cm dishes per culture condition. Translation was arrested by incubating cells with 100 μg/ml cycloheximide (CHX, Sigma # C4859) for 10 min at 37°C. After washing cells on ice with PBS supplemented with 100 μg/ml CHX, cells were lysed in buffer (10 mM Tris–HCl (pH 7.4), 5 mM $MgCl_2$, 100 mM KCl, 1% Triton X-100, 2 mM DTT, 500 U/ml RNase inhibitor, 100 μg/ml CHX, and protease inhibitors). Cell lysates were then sheared gently four times with a 26-gauge needle. Lysates were then collected after centrifugation at $1,300 × g$ for 10 min. Lysates were layered onto 10–50% sucrose gradients and centrifuged in an SW-41Ti rotor at 36,000 r.p.m. for 2 h. Gradients were fractionated using a BioComp Gradient Station fractionator, and absorbance at 254 nm was monitored to obtain the polysome profile. Polysome:monosome (P:M) ratios were derived by integrating the area under the respective peaks.

## Small RNA sequencing from polysomes

Prior to polysome profiling, cells were translationally arrested by treating with CHX for 30 min. Lysates made from various states of differentiation (*stem: Wnt3a + LIF- and LIF-treated; differentiating: RA- and Wnt3a + RA-treated*) were ultracentrifuged on a 15–45% sucrose gradient at 39,000 rpm to separate the ribosomal fractions. The individual ribosomal subunits (40S, 60S), monosomes (80S), and polysomes were fractionated based on the profile generated by the absorbance of the gradient at 254 nm. The RNA was precipitated from the sucrose gradient collected by ethanol precipitation. The pellets were dissolved in TRIzol, and RNA was isolated from the pellets. Small RNA libraries were prepared from isolated RNA using TruSeq Small RNA Prep Kit (RS200012, Illumina). For statistics, samples treated with Wnt3a+LIF and LIF were grouped as proxies for stem, and RA- and Wnt3a+RA-treated (absence of LIF) as proxies for differentiating state. The data were analyzed as described in Methods Section "Identification and characterization of tsRNAs".

We used Wilcox's test [46] to calculate significance. The change in the tsRNA profiles was calculated using change point analysis (R package: changepoint) [61].

## tsRNA-mediated mRNA pulldown library and analysis

The paired-end sequencing reads were adapter-trimmed using Trimmomatic [49] and aligned to mouse (mm10) genome using Tophat v2.0.9 (-p 10 -o./-i 50 -I 250000) [62]. Around 92–95% of reads mapped back to reference genome. We used HTSeq-count v0.6.0 [63] (-f bam –r name –s no –m union -o) to obtain raw read numbers for all the RefSeq transcripts. Gene body coverage analysis, for all samples, showed reads covering the entire transcript and not from one specific region. The count data were normalized using DESeq. We used a cut-off of DESeq normalized value > 200 in at least one of the four samples (Scr1 and tsRNA pulldown in LIF and RA). We further selected transcripts that showed tsRNA/Scr1 fold-change ratio to be more than twofold. To identify transcripts that significantly associate to tsRNA in comparison with Scr1, we used Fisher exact test [64] [fisher.test (x,alternative="two.sided",simulate.*P*.value = FALSE,B = 2000)]. *P*-value obtained from Fisher exact test was corrected using *Bonferroni* method (stats package in R) [14]. Transcripts with adjusted *P*-value < 0.05 were considered as significant. Either in LIF or in RA, 3,664 transcripts showed significant association in tsRNA pulldown compared with their respective control (Scr1; Dataset EV5).

Further, to identify cell-state-specific tsRNA-associated transcripts we pruned this list based on tsRNA_RA/tsRNA_LIF fold-change values. A cut-off of twofold was used to allocate transcripts as either RA-enriched (420 transcripts) or LIF-enriched (28 transcripts). We mined publicly available transcriptome data [65] and classified transcripts that were more than twofold upregulated after RA treatment as "differentiation-responsive genes" and more than twofold downregulated as "pluripotency-associated genes". Genes that showed tsRNA/Scr1 was > 2-fold in both the LIF and RA conditions and similar enrichments in both the conditions were categorized as "Ubiquitous". Subsequently, we correlated our tsRNA pulldown data with these two classes of gene sets to pinpoint molecular players that tsRNA might most possibly be regulating and interacting. The GO analyses for different categories were done using g:profiler (http://biit.cs.ut.ee/gprofiler/; Dataset EV6) [53].

## tsRNA target interaction assays

Equal amounts of protein lysates (500 μg in 500 μl) were taken for the pulldown. While one part was considered "normal lysate", total RNA was isolated from the other part using TRIzol. The isolated RNA was resuspended in 500 μl of lysis buffer (this was deemed as lysate devoid of proteins). 3 μg of biotin-tagged tsGlnCTG was added to lysates with and without proteins, and the tsRNA-interacting RNAs were isolated as described in the section "Biotinylated RNA mimics pulldown". 1 μl of RNA was taken from both conditions and converted to cDNA using SuperScript III RT (Invitrogen). Differential enrichments of tsRNA interacting transcripts were calculated between lysates devoid of proteins and normal lysates by qPCR analysis. The experimental design is depicted in Fig EV4H.

## Western blotting

After PAGE and PVDF or nitrocellulose membrane transfer using standard laboratory protocols, antibodies against IMP1/IGF2BP1 (Cell Signaling Technologies D33A2, #8482S) and β-actin (Abcam 8226, ab8226) were used to probe for target proteins. IR Dye 680RD Goat anti-Rabbit (LiCor 926-68071, Lot# C31205-05) and IR Dye 800CW Goat anti-Mouse (LiCor 926-32210, Lot# C40213-01) were then used as secondary probes against primary antibodies. Immunoblots were imaged with the LiCor Odyssey® CLx Imaging System.

## siRNA knockdown

Three Silencer® Select siRNAs against each gene target were purchased from Ambion, Thermo Fisher Scientific. Information on the siRNAs may be found in Dataset EV8. 10 nM of pool of three different siRNAs was used to knockdown each target gene.

## UV-RNA immunoprecipitation assay

2-day LIF- or RA-treated mouse ESCs were UV-crosslinked at 450 mJ/cm$^2$. Cell extracts were made using a lysis buffer (100 mM Tris pH 7.4, 150 mM NaCl, 1% NP40, RNase inhibitor, protease inhibitor cocktail, and 1 mM DTT). 1 mg of cell extract was taken for immunoprecipitation following which extracts were pre-cleared with 50 μl protein A/G resin (Thermo Scientific, 53132) for an hour at 4°C. Pre-cleared lysates were incubated with Igf2bp1 antibody (Cell Signaling Technologies, 2852S) or IgG (Cell Signaling Technologies, 3900S) control overnight at 4°C. Antibody was immuno-precipitated using protein A/G-coated resin and subsequently washed four times with lysis buffer. To release the RNA from the protein complex, the pulldown sample was treated with 5 mg/ml proteinase K at 37°C for 10 min. The RNA was extracted using TRIzol. Small RNA libraries were made using the isolated RNA as described using TruSeq Small RNA Prep Kit. The libraries were sequenced and analyzed as discussed previously. For qPCR analysis of target genes, 1 μl of the suspended RNA was reverse-transcribed with SuperScript III RT (Invitrogen) using random hexamers or tsRNA-specific RT primer. qPCR of the reverse-transcribed cDNA was carried out on Bio-Rad CFX384 Touch machine using Fermentas Power SYBR Green Master Mix and respective gene-specific or tsRNA-specific primers.

## Transcript association polysome fractions

Fractions (non-translating, 80S and polysome) from polysome profiling were collected based on the profile. The RNA was precipitated from the collected fraction using ethanol precipitation. The pellet was resuspended in TRIzol to extract RNA. 1 μg of RNA was taken for reverse transcription using SuperScript III RT (Invitrogen). The reverse-transcribed RNA was taken for qPCR analysis for target genes between LIF- vs. RA-treated mESCs. GAPDH was used as the internal control, and the fold changes for each fraction in RA-treated mESCs were calculated with respect to the LIF fractions. qPCR was carried out on Bio-Rad CFX384 Touch machine using Fermentas Power SYBR Green Master Mix and respective gene-specific primers.

### *In vitro* tsRNA-Igf2bp1-binding assays

Prior to binding assays, biotin-tagged tsGlnCTG was incubated at 95°C with or without antisense oligo (10 molar excess) in RNA structure buffer (10 mM Tris pH 7.0, 0.1 M KCl, 10 mM $MgCl_2$), heated to 95°C, and gradually cooled to room temperature. Binding assays were carried out in 100 μl reaction mixture (5 mM Tris pH 7.4, 2.5 mM EDTA pH 8, 2 nM DTT, 5% glycerol, 0.1 mg/ml BSA, RNase inhibitor). 400 nM of tsGlnCTG was incubated with increasing concentrations (20, 40, 80, 200, and 400 nM) of recombinant Igf2bp1 (TP316226, Origene) for 30 min at 30°C. 100 μl of Dynabeads M-280 Streptavidin™ (11206D, Invitrogen) was added to the reaction and incubated at room temperature for 30 min. The beads were magnetized to separate the bound tsGlnCTG. The beads were washed thrice with washing buffer (20 mM Tris–HCl pH 7.5, 150 mM NaCl, 1.8 mM $MgCl_2$, 0.5% NP40). The beads were resuspended in 30 μl of 1× Laemmli buffer and heated. The supernatant was loaded on a 10% polyacrylamide gel and transferred onto nitrocellulose membrane for Western analysis. The membrane was probed with Igf2bp1 antibody (Cell Signaling Technologies D33A2, #8482S) followed by HRP-secondary antibody. The chemiluminescence was visualized using ImageQuant LAS 4000 machine (GE healthcare). The band intensities of tGlnCTG-pulled Igf2bp1 were calculated (using ImageJ) and normalized to the intensity 400 nM of Igf2bp1 as percentages. The relative band intensities were plotted to get the binding affinity (Kd).

### Myc-Luc reporter activity assay

The HCT116 Myc-Luc reporter cells were reverse-transfected in 384-well plates. Transfection mix consisting of transfectants, Lipofectamine 2000 (used according to manufacturer's recommendations), and Opti-MEM was pipetted into respective wells (20 μl per well). 5,500 cells in 30 μl of 16.7% FBS-supplemented DMEM media were then seeded in each well. After 72 h, cell viability was assayed for using PrestoBlue reagent (Thermo Fisher Scientific A13262), followed by Steady-Glo Luciferase Assay (Promega E2550) for Myc activity. Fluorescence and luminescence readings were obtained with the Tecan Infinite M1000 PRO Multimode Microplate Reader.

## Data availability

The sequencing data generated for this study have been deposited in Sequence Read Archives, National Center for Biotechnology Information (SRA). The raw sequences can be accessed with the project ID SRP079660.

**Expanded View** for this article is available online.

### Acknowledgements

We would like to extend our gratitude to Drs Wan Yue (GIS), Francesc Xavier Roca Castella (NTU), Apurva Sarin (inStem), Vinoth Kumar (NCBS), Tina Mukerjee (inStem) and Sunil Laxman (inStem) for comments and critical review of the manuscript. We are thankful for technical support from the following: the Centre for High-throughput Phenomics (CHiP-GIS) Singapore, for RNAi-based functional validation of candidate proteins identified in tsRNA pulldown screen; GIS-NGS facility, Next Generation Genomics Facility (NGGF) at inStem and Genotypic Technology, Bangalore, for sequencing; the A*STAR SiGN Common FACS Facility, and Central Imaging and Flow Facility (CIFF, Bangalore) for FACS-based applications at inStem, Bangalore. This work was supported by the following grants: DST Swarnajayanti fund (DST/SJF/LSA-02/2015-16) and Wellcome-DBT India Alliance Intermediate Fellowship (500160/Z/09/Z) to DP; DBT Grant (BT/PR8655/AGR/36/759/2013) to RD, SR; inStem core funds through the DBT to SR and DP; and Agency for Science, Technology and Research (A*STAR)-GIS core funds to RD. DGRY was supported by the A*STAR Graduate Academy (AGA). VL was supported by CSIR SRF fellowship.

## Author contributions

RD, DP, and SR conceived and designed the study. SK and DGRY performed experiments. VL, JLYK, and DP performed all computational analyses. VT and PS helped with Northern hybridization experiments. JEP and SKS contributed critically to mass spectrometry; JKC contributed to experiments on HMEC lines; and JLL and MJSL helped with cell culture and Myc reporter assays. AG contributed intellectually to tsRNA-binding assays. DP, SR, RD, DGRY, and SK wrote the manuscript with extensive input from all authors.

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
