## [Review Process File · EMBO Reports]

Dynamic expression of tRNA-derived small RNAs define cellular states

Srikar KRISHNA, Daniel GR YIM, Vairavan LAKSHMANAN, Varsha TIRUMALAI, Judice LY KOH, Jung Eun PARK, Jit Kong CHEONG, Joo Leng LOW, Michelle JS LIM, Siu Kwan SZE, Padubidri SHIVAPRASAD, Akash GULYANI, Srikala RAGHAVAN, Dasaradhi PALAKODETI, Ramanuj DASGUPTA

Review timeline:

Submission date:	25 January 2019
Editorial Decision:	26 February 2019
Revision received:	3 April 2019
Editorial Decision:	2 May 2019
Revision received:	7 May 2019
Accepted:	15 May 2019

Editor: Deniz Senyilmaz Tiebe

Transaction Report:

1st Editorial Decision

26 February 2019

Thank you for submitting your manuscript for consideration by EMBO Reports. It has now been seen by three referees whose comments are shown below.

As you can see, all referees express interest in the proposed function of tRNA-derived small RNAs in mESC differentiation. However, they also raise concerns that need to be addressed in full before we can consider publication of the manuscript here.

Given these constructive comments, I would like to invite you to revise your manuscript with the understanding that the referee must be fully addressed and their suggestions taken on board. In particular, description of the methodology and the depth of analysis must be significantly improved. Please address all referee concerns in a complete point-by-point response. Acceptance of the manuscript will depend on a positive outcome of a second round of review. It is EMBO Reports policy to allow a single round of revision only and acceptance or rejection of the manuscript will therefore depend on the completeness of your responses included in the next, final version of the manuscript.

Supplementary/additional data: The Expanded View format, which will be displayed in the main HTML of the paper in a collapsible format, has replaced the Supplementary information. You can submit up to 5 images as Expanded View. Please follow the nomenclature Figure EV1, Figure EV2

etc. The figure legend for these should be included in the main manuscript document file in a section called Expanded View Figure Legends after the main Figure Legends section. Additional Supplementary material should be supplied as a single pdf labeled Appendix. The Appendix includes a table of content on the first page with page numbers, all figures and their legends. Please follow the nomenclature Appendix Figure Sx throughout the text and also label the figures according to this nomenclature. For more details please refer to our guide to authors.

When preparing your letter of response to the referees' comments, please bear in mind that this will form part of the Review Process File, and will therefore be available online to the community. For more details on our Transparent Editorial Process, please visit our website:
http://emboj.embopress.org/about#Transparent_Process

Regarding data quantification, please ensure to specify the name of the statistical test used to generate error bars and P values, the number (n) of independent experiments underlying each data point (not replicate measures of one sample), and the test used to calculate p-values in each figure legend. Discussion of statistical methodology can be reported in the materials and methods section, but figure legends should contain a basic description of n, P and the test applied. Please also include scale bars in all microscopy images.

We now strongly encourage the publication of original source data with the aim of making primary data more accessible and transparent to the reader. The source data will be published in a separate source data file online along with the accepted manuscript and will be linked to the relevant figure. If you would like to use this opportunity, please submit the source data (for example scans of entire gels or blots, data points of graphs in an excel sheet, additional images, etc.) of your key experiments together with the revised manuscript. Please include size markers for scans of entire gels, label the scans with figure and panel number, and send one PDF file per figure.

- a complete author checklist, which you can download from our author guidelines (<http://embor.embopress.org/authorguide#revision>). Please insert page numbers in the checklist to indicate where the requested information can be found.
 - a letter detailing your responses to the referee comments in Word format (.doc)
 - a Microsoft Word file (.doc) of the revised manuscript text
 - editable TIFF or EPS-formatted figure files in high resolution
- (In order to avoid delays later in the publication process please check our figure guidelines before preparing the figures for your manuscript:
http://www.embopress.org/sites/default/files/EMBOPress_Figure_Guidelines_061115.pdf)
- a separate PDF file of any Supplementary information (in its final format)
 - all corresponding authors are required to provide an ORCID ID for their name. Please find instructions on how to link your ORCID ID to your account in our manuscript tracking system in our Author guidelines (<http://embor.embopress.org/authorguide>).

As part of the EMBO publication's Transparent Editorial Process, EMBO reports publishes online a Review Process File to accompany accepted manuscripts. This File will be published in conjunction with your paper and will include the referee reports, your point-by-point response and all pertinent correspondence relating to the manuscript.

I look forward to seeing a revised version of your manuscript when it is ready. Please let me know if you have questions or comments regarding the revision.

REFeree REPORTS

Referee #1:

This manuscript aims to characterize novel small non-coding RNAs that might play a role in cell differentiation. The authors identified 5' half tRNA fragments that are increased in mouse ESC RA (retinoic acid)-induced differentiation model by small RNA-sequencing and validated by Northern blot. Knocking down tRFs in differentiated ESC increased cell pluripotency gene expression, suggesting a role of tRFs in defining cell identity. This is a novel observation and adds one more function about the less-studied tRNA fragments. The authors explored diverse possible mechanisms through which these tRNA fragments/halves could exert their functions in the process of differentiation by multiple omics approaches. Most interestingly, they found tRF binding with IGF2BP1 (IMP1) protein, a protein important for stem cell function and known to bind/stabilize c-Myc mRNA. Their data suggests tRF sequesters Igf2bp1 from the cMyc mRNA-protein complex, which in turn results in decreased expression and translation of cMyc mRNA thereby driving and/or facilitating the maintenance of the differentiated state in RA-treated mESCs. Overall I find this a comprehensive study and would bring attention to the novel biological role and mechanism of action of tRNA fragments.

Minor:

1. Fig. 1E font is too small to see clearly; if space is limited, Fig 1D might be moved to supplementary. Besides, what's the source of the tRNA folding in Fig 1E? - it should be cited in the Fig legend or methods section.
2. It is not clear what the authors want to show in Fig. 2A. The pie chart is difficult to read quantitatively and not clear what is it normalized to. The tRF naming on the right is also confusing and hard to compare with the Northern blot results.
3. It is not clear how the authors performed the tRF qPCR with total RNA (which will inevitably have full-length tRNA and interfere with the quantification). Maybe the authors considered this possibility, if so, the precautions should be clearly stated in the text and methods section. If it is not clear how tRNA will affect the results, I will suggest moving all qPCR results to supplementary and just leave the Northern blot results in the main figures. The authors may consider plotting tRNA abundance (with a dashed line) in the same plot as Fig. 2C' as a good internal control.
4. Fig. 3C the figure legend inside the figure is not clear enough.
5. Fig. 4A title "a common protein" doesn't reflect the venn diagram here, unless the authors actually want to emphasize IGF2BP1, in that case they should label IGF2BP1 in the common set. The tRF-IGF2BP1 interaction seems to be the highlight of the story but is diluted by some other results. I would suggest labeling IGF2BP1 in Fig. 4A and moving Fig. 4C&D to supplementary.
6. Fig. 5A&B, error bar is missing.
7. Page 7, "We further validated the interaction of 5'tsRNA with the translating units by sequencing fractions obtained from polysome profiling in mESCs grown under stem or differentiating conditions" miss the reference to the figure.
8. Figure 6, I think "Angiogenin" should be removed from this illustration and will not affect the impact of the paper, since the authors are not clear if this process is Angiogenin-dependent.

Referee #2:

The manuscript Krishna et al entitled "Dynamic expression of tRNA-derived small RNAs define cellular stress" describes how transfer RNA-derived small RNAs can modulate stability and translation of c-Myc mRNA in mESCs, providing a potential mechanism linking tRNA fragments to cell proliferation.

The conclusions are potentially important. However, several of the experiments are minimally described, making them difficult to interpret. I recommend major revision of the MS, especially

correction or deeper description of the high-throughput data analysis. The MS would also benefit from focusing more on the well supported mechanistic conclusions.

Major points:

1: It is impossible to evaluate the high-throughput part of the MS due to the limited description of the mapping methodology. tRNAs molecules are encoded by multiple genes and have to be added "manually" to certain databases. Authors should provide following information:

1.1 Were reads mapped allowing multiple locations?

1.2 What was the minimal length of mapped reads in bowtie runs?

1.3 Does the database contain only mature tRNAs or also precursors?

1.4 What percent of reads mapped to rRNA?

2: Fig. 1 and S1: How were the data for tRNAs normalized. The fraction of 30 nt RNAs may or may not reflect changes in absolute abundance per cell. This is also a potential issue for Fig. S2. Genomic and tRNA databases should be combined and percent of reads mapped to tRNA should be calculated using combined database. It would be better to use some alternative and bias aware quantification method as a control for quantification (for example using Salmon aligner).

3: Fig 1D-E - The authors show only a few profiles. A more advanced analysis is needed to withdraw conclusion that "most tsRNAs were derived from the 5' halves". Fig S1I is most likely this kind of analysis. Descriptions of polysome profiling in mESCs is not referred to a figure. Is it Fig S4C? If so, a better representation of the data should be provided.

4: Fig. 3 - The use of ASOs to block putative tsRNA interactions is a significant experiment in establishing a function for tsRNA, but is very minimally described. Which ASOs were used to block tsRNAs? Was this only anti-tsGlnCTG - or a panel of ASOs? If the former - are the authors proposing that this is the sole tsRNA responsible for the proliferation regulation? If the latter, what fraction of the tsRNA population was affected?

5: Fig 4B - The relevance of ribosomal proteins found in this analysis is unclear. The authors report that tRFs are associated with translating ribosomes using polysome profiling. Most of the proteins reported in this figure are ribosomal proteins, but some are described as connected to proliferation whereas other to differentiation. It does not appear that ribosome heterogeneity can explain this result. Other explanations should be considered (interaction site?) or a higher stringency of statistical test could be used to eliminate most of ribosomal proteins. Maybe a SILAC based method could be implemented.

6: The results are discussed to large extent within the "Result" section whereas the Discussion covers elements not covered by the MS, including tRF biogenesis and regulation.

Minor points:

7: One Sentence Summary - if this is intended as running title then should be changed

8: Two different abbreviations are used: tsRNAs and tRFs - could this be changed? Are 5'-tRFs and 5'-tsRNAs equivalent?

9: LIF - name of LIF protein is not explained when used for the first time.

10: Is there a correlation between amount of tRNA halves recovered and abundance of tRNA and/or gene copy number?

11: Fig. 2C - could include tsGluGCC

12: For direct identification of tRFs-mRNA interaction some previously published datasets could be used (PARIS, LIGR or CLASH methods)

13: "...angiogenin. the. Additionally,..." - to be corrected

14: "spatial positioning of 5'-tsRNAs on the ribosome" could be easily validated by analysis of the position of found ribosomal proteins on the ribosome structure. See also Major point 5.

Referee #3:

The manuscript submitted by Krishna et al. reports on upregulated tRNA 5'-halves (called 5'-tsRNAs in this submission) in mouse embryonic stem cells (mESC) upon retinoic acid-induced differentiation. The authors provide primarily high-throughput data suggesting a role of a subset of these tRNA 5'-halves in modulating protein biosynthesis as well as mRNA stability of certain key factors such as c-Myc.

The topic of tRNA-fragments and their involvement in regulating gene expression is timely, relevant and interesting. However, the manuscript submitted suffers from some over-interpretations and the lack of crucial control experiments.

1) The authors state that they have studied the role of several tRNA 5'-halves by overexpressing them. This is misleading since from the Methods chapter it is clear that they used transfection to introduce synthetic RNA into mESC. Strictly speaking this is not "overexpression". It is furthermore pivotal to directly show the extent of increased tRNA 5'-half levels inside the cell (e.g. by northern blot or qPCR). Without this information the data shown in Fig. S3 cannot be interpreted.

2) The data shown by the authors point into two possible directions: (1) the tRNA 5'-halves bind the ribosome and polysomes and modulate translation or (2) tRNA 5'-halves associate with specific mRNAs (likely via RNA-binding proteins) and regulate the stability and/or translatability of these messages. The authors need to more rigorously discuss these two options because they somewhat appear mutual exclusive. The *in vitro* translation assays shown in Suppl. Fig. S4B revealed translation inhibition of both their two reporter transcripts (luciferase and GFP). Thus it seems that the regulatory role of the tRNA 5'-halves on translation is mRNA sequence independent since neither luciferase nor GFP can be genuine targets of tRFs in mouse cells.

3) related to point #2: the potential mRNA-specific effect of tRNA 5'-halves on mRNA stability seems to be remarkably analogous to miRNA action. Indeed tRNA fragments have been shown in the past to function as mi/siRNAs. Can the authors indeed exclude the possibility that their tRNA 5'-halves associate with Argonaut and function in an RNAi-like fashion? Since they made pull-downs of the tRNA halves and have analyzed the interacting proteins, they should actually have the data. If so this should be more explicitly discussed in the text.

4) also related to #2: the authors show that the tRNA fragments under investigation seem to be located in ribosome and polysome fractions on their density gradients. First, the authors should show the polysome gradient profile in order for the reader to assess the quality of their particle separation. Second, there is recent literature on the regulatory role of ribosome-bound small non-coding RNAs. The authors might want to consider discussing their findings in respect to the current state in the field.

5) Fig. 2C: I strongly suggest replacing the northern blots shown by those from Supplementary Fig. S2A. When studying tRNA-fragments it is always important and reassuring to also see the signal of the full length tRNA molecules on the same blot.

6) The authors performed small RNA-seq before and after T4 PNK treatment in order to check the status of the 3'-ends of their tRNA halves (Fig. S1D). They conclude that, based on the obtained reads, Angiogenin might not be the main processing enzyme. However there is a 30% increase of tRNA-fragment reads upon T4 PNK treatment which is reminiscent of Angiogenin involvement. In fact the better and more direct way to ascertain an Angiogenin involvement in tRNA halves production would be RNAi against Angiogenin followed by small RNA-seq.

7) in general: the observed effects involving tRNA 5' halves (e.g. Fig. 1B, Fig. 3A, Fig 5B, Fig 5F,

Fig. S3A, B) are quite modest. Therefore a more careful interpretation of the data and more toned-down conclusions are appropriate.

1st Revision - authors' response

3 April 2019

Referee #1:

This manuscript aims to characterize novel small non-coding RNAs that might play a role in cell differentiation. The authors identified 5' half tRNA fragments that are increased in mouse ESC RA (retinoic acid)-induced differentiation model by small RNA-sequencing and validated by Northern blot. Knocking down tRFs in differentiated ESC increased cell pluripotency gene expression, suggesting a role of tRFs in defining cell identity. This is a novel observation and adds one more function about the less-studied tRNA fragments. The authors explored diverse possible mechanisms through which these tRNA fragments/halves could exert their functions in the process of differentiation by multiple omics approaches. Most interestingly, they found tRF binding with IGF2BP1 (IMP1) protein, a protein important for stem cell function and known to bind/stabilize c-Myc mRNA. Their data suggests tRF sequesters Igf2bp1 from the cMyc mRNA-protein complex, which in turn results in decreased expression and translation of cMyc mRNA thereby driving and/or facilitating the maintenance of the differentiated state in RA-treated mESCs. Overall I find this a comprehensive study and would bring attention to the novel biological role and mechanism of action of tRNA fragments.

We thank this reviewer for their encouraging comments and are delighted to know that they find our study “comprehensive”. We have made every attempt to address all of the questions/suggestions raised by this reviewer.

Minor:

1. Fig. 1E font is too small to see clearly; if space is limited, Fig 1D might be moved to supplementary. Besides, what's the source of the tRNA folding in Fig 1E? - it should be cited in the Fig legend or methods section.

We have now changed the font sizes in Fig. 1E.

We used VARNA: Visualization Applet for RNA (Darty *et al.*, 2009) to plot the tRNA folding. This has been described in the methods section “Identification and characterization of tsRNAs”.

“We used VARNAv3-93 (Darty, Denise and Ponty, 2009) for obtaining secondary structures of tRNA and we used the coverage values obtained to plot the tRNA heatmaps.” (Page No. 15)

2. It is not clear what the authors want to show in Fig. 2A. The pie chart is difficult to read quantitatively and not clear what is it normalized to. The tRF naming on the right is also confusing and hard to compare with the Northern blot results.

The pie chart depicts the percentage representation of each species of tsRNAs with respect to the total tsRNA population of that particular sample. The data suggests that not all tRNAs are processed into tsRNAs, the total tsRNAs population is contributed by the processing of 5-6 tRNAs. Further, we have now used consistent tsRNA nomenclature to avoid confusion and added percentages to the pie charts for more clarity.

3. It is not clear how the authors performed the tRF qPCR with total RNA (which will inevitably have full-length tRNA and interfere with the quantification). Maybe the authors considered this possibility, if so, the precautions should be clearly stated in the text and methods section. If it is not clear how tRNA will affect the results, I will suggest moving all qPCR results to supplementary and just leave the Northern blot results in the main figures. The authors may consider plotting tRNA abundance (with a dashed line) in the same plot as Fig. 2C' as a good internal control.

As per the reviewer's suggestion, we have moved the northern blots to the main figure (Fig 2B) and the QPCRs to the supplementary figure (Fig S2A). We have also added the tRNA abundance in the same plot as dashed lines (Fig. 2C').

Regarding the specificity issue raised on our QPCRs, we use a well-established protocol similar to the ones used to quantitate miRNAs as well as tsRNAs (Varkonyi-Gasic and Hellens, 2011; Peng *et al.*, 2012; Gautam *et al.*, 2016; Marcial-Quino *et al.*, 2016). The use of a stem-loop adapter in the RT Primer hinders the binding to the mature tRNA. Further, to ascertain the specificity of the QPCR, the amplified products were run on a gel. We observed only a single band corresponding to the expected size of the tsRNA (~75bp including adapters). We did not observe any band sizes corresponding to the expected mature tRNA (~115 bp). We have now added this data to the supplementary figure (Fig. S2A').

4. Fig. 3C the figure legend inside the figure is not clear enough.

As per reviewers suggestion, we have increased the fonts of the legends inside the figure.

5. Fig. 4A title "a common protein" doesn't reflect the venn diagram here, unless the authors actually want to emphasize IGF2BP1, in that case they should label IGF2BP1 in the common set. The tRF-IGF2BP1 interaction seems to be the highlight of the story but is diluted by some other results. I would suggest labeling IGF2BP1 in Fig. 4A and moving Fig. 4C&D to supplementary.

The list of 109 "common" proteins includes IGF2BP1 as one of the most abundantly interacting protein, quantitated in the supplementary table S4. To avoid confusion, we have now labelled IGF2BP1 in the venn diagram (Fig. 4A).

Respectfully, we believe that Fig 4C and D should remain part of the main figure as it points to a key finding regarding the specific interaction of tsRNA with the transcripts under stem vs differentiating conditions.

6. Fig. 5A&B, error bar is missing.

We have now added the error bars to the Figure 5A. With regards to Figure 5B, the replicate of this experiment is plotted in Supplementary figure S5F. Due to variable efficiencies in pulldown, plotting error bars would dilute the effect of the experiment. Nonetheless, the trends observed in cMyc binding are the same in both replicates.

7. Page 7, "We further validated the interaction of 5'tsRNA with the translating units by sequencing fractions obtained from polysome profiling in mESCs grown under stem or differentiating conditions" miss the reference to the figure.

We thank the reviewer for pointing this out. We have now added the Figure reference (now Fig. S4C') to the above mentioned line.

8. Figure 6, I think "Angiogenin" should be removed from this illustration and will not affect the impact of the paper, since the authors are not clear if this process is Angiogenin-dependent.

As per the reviewer's suggestion we have now removed angiogenin from the figure.

Referee #2:

The manuscript Krishna et al entitled "Dynamic expression of tRNA-derived small RNAs define cellular stress" describes how transfer RNA-derived small RNAs can modulate stability and translation of c-Myc mRNA in mESCs, providing a potential mechanism linking tRNA fragments to cell proliferation.

The conclusions are potentially important. However, several of the experiments are minimally described, making them difficult to interpret. I recommend major revision of the MS, especially correction or deeper description of the high-throughput data analysis. The MS would also benefit from focusing more on the well supported mechanistic conclusions.

We thank the reviewer for insightful and constructive comments and finding our conclusions potentially important. We have made extensive changes to the manuscript which include a “deeper description of the high-throughput data analysis” as suggested by the reviewer. We hope we have addressed the queries satisfactorily.

Major points:

1: It is impossible to evaluate the high-throughput part of the MS due to the limited description of the mapping methodology. tRNAs molecules are encoded by multiple genes and have to be added "manually" to certain databases. Authors should provide following information:

1.1 Were reads mapped allowing multiple locations?

The GtRNAdb consists of tRNAs from multiple locations (multiple genes) in the genome. To avoid multimapping we clustered the tRNAs that share the same sequences in the first 50 nt for every tRNA isotype. In supplementary table S2 the clustered IDs along with the parent tRNA IDs are mentioned. This has been explained in the revised methods section “Identification and characterization of tsRNAs” (Page No. 15).

“We used mm10 (Genome and Consortium, 2002) and hg19 (Consortium, 2001) as the reference genomes (<http://genome.ucsc.edu/>) for mouse and human samples respectively. From the sequencing reads, we trimmed TruSeq small RNA adapters using customized perl script and cutadapt program (-f fasta -b TGGGAATTCTCGGGTGCCAAGG -O 5 -m 6 -o) (Martin, 2011). The adapter trimmed reads were aligned to rRNA and unaligned reads were taken for further analysis. For analysis, we considered reads ranging between 18-35 nucleotides and mapped these to genome and the tRNA database obtained from GtRNAdb (Chan and Lowe, 2009) using bowtie v1.1.2 (-f -v 2 -p 20 -un) (Langmead *et al.*, 2009). We used 2 mismatches as a constant parameter for all the alignments done in this study, as we did not observe much deviation in our results with varying mismatches ranging from 0-2. Reads, which are 30-35 nucleotides in length and specifically maps only to tRNA database, were considered as tsRNAs. Since multiple loci on the genome code for the same tRNA isotype, and to avoid multi-mapping problem (same reads mapping to multiple locations), we clustered the tRNA based on sequence similarity of the first 50 nucleotides (468 tRNAs were clustered to 226 unique tRNA sequences). We have provided the cluster information in Supplementary Table S2. We did not consider 18-24 mer reads as tsRNAs because only negligible reads mapped to tRNA database. To calculate the percentage tsRNA reads that mapped to genome, we calculated the ratio of reads of a particular size that mapped to tRNA to the total reads that mapped to the genome (Fig. 1B). Further, we calculated per base tRNA coverage using the following formula: *Per base coverage* = (number of reads aligned to particular base of tRNA/total number of reads mapped to all the tRNA in that sample) which is plotted in Figure 1D.

We used VARNAv3-93 (Darty, Denise and Ponty, 2009) for obtaining secondary structures of tRNA and we used the coverage values obtained to plot the tRNA heatmaps. We used DESeq (Anders and Huber, 2010) for normalization of the sequencing data and identification of differentially expressed 5'-tsRNAs (adj. *P*-value < 0.05). All the statistical tests were done in R (R Development Core Team, 2011). We used customized perl script for all the analysis used in this study. We used R ggplot2 (Wickham, 2009) library for plotting. The raw numbers and mapping percentage for all the sequenced samples are given in Table S1.”

1.2 What was the minimal length of mapped reads in bowtie runs?

In this study, we prepared the small RNA libraries using TruSeq small RNA prep kit (Illumina). According to the manufacturers protocol, we size selected the prepared library between 140-165 nt (a ladder provided by Illumina). This size range selects for the all small RNAs ranging from 18-35 nts. We considered only the reads spanning 18-35 nt for analysis and hence the minimal length that mapped to the genome would be and 18 nucleotides. Please refer methods section “Identification and Characterization of tsRNAs” (page number 16).

“For analysis, we considered reads ranging between 18-35 nucleotides and mapped these to genome and the tRNA database obtained from GtRNAdb (Chan and Lowe, 2009) using bowtie v1.1.2 (-f -v 2 -p 20 -un) (Langmead *et al.*, 2009).”

1.3 Does the database contain only mature tRNAs or also precursors?

The GtRNADB database used for mapping reads to tRNA contains only the mature tRNAs (Chan and Lowe, 2009).

1.4 What percent of reads mapped to rRNA?

The percentage of reads that map to rRNA is 24-34% (in Wnt3a+LIF, LIF- and RA-treated conditions). We have provided this information in the supplementary table S1.

2: Fig. 1 and S1: How were the data for tRNAs normalized. The fraction of 30 nt RNAs may or may not reflect changes in absolute abundance per cell. This is also a potential issue for Fig. S2.

tsRNA normalization in Figure 1 and S1:

The sequencing data was normalised using the widely used DESeq algorithm which normalises the data based on the size factor. The size factor is calculated as the median of the depth ratios per sample. In addition to this, we also normalized the tsRNA reads to the total number of reads that mapped to the genome. From the normalized reads, we calculated the percentage abundance of tsRNAs. We agree with the reviewer that the fraction of tsRNAs may not reflect absolute levels, and hence we have used relative measure for analysis.

Most importantly, we have substantiated the sequencing data with extensive validations through Northern hybridization and small RNA QPCR.

We have provided an elaborate description in the methods section “Identification and characterization of tsRNAs” on page no 15.

“We used the well-established DESeq algorithm (Anders and Huber, 2010) for normalization of the sequencing data and identification of differentially expressed 5'-tsRNAs (adj. *P*-value < 0.05).”

“To calculate the percentage tsRNA reads that mapped to genome, we calculated the ratio of reads of a particular size that mapped to tRNA to the total reads that mapped to the genome (Fig. 1B).”

tRNA and tsRNA Normalisations for the Northern blots in Fig 2:

We like to point out that the northern blots in Fig. S2 has been moved to Fig. 2. Equal RNA was loaded for the Northern blot and the band intensities of the tRNA or tsRNA were normalised to the band intensities of U6snRNA (loading control) for every sample. It is interesting to note that while the expression of tRNAs remained unchanged, the tsRNAs expression was more dynamic over the timepoints tested.

Genomic and tRNA databases should be combined and percent of reads mapped to tRNA should be calculated using combined database. It would be better to use some alternative and bias aware quantification method as a control for quantification (for example using Salmon aligner).

The reads that mapped to the tRNA database (GtRNADB) was a subset of the reads that mapped to the genome. This database considers multiple loci that encodes for a tRNA (total 468 loci).

For relative quantitation, the percentage of reads that mapped to tRNAs were calculated as the reads that mapped to tRNAs divided by the total reads that mapped to the genome (for example as seen in figure 1B).

As suggested by the reviewer, we used Salmon aligner to map the reads to tRNA. We see a high correlation between the data obtained from Bowtie vs Salmon aligner ($R^2 = 0.99$). Please refer to the figure below. This gives us high confidence in our mapping and we thank the reviewer for this suggestion.

3: Fig 1D-E - The authors show only a few profiles. A more advanced analysis is needed to withdraw conclusion that "most tsRNAs were derived from the 5' halves". Fig S1I is most likely this kind of analysis. Descriptions of polysome profiling in mESCs is not referred to a figure. Is it Fig S4C? If so, a better representation of the data should be provided.

For all tRNA lengths, the perbase coverage was calculated and plotted in Figure 1D. Figure 1D takes into consideration all the tRNAs and the number of reads that map at every nucleotide of the parent tRNA. As seen in Fig1D most reads map predominantly to the 5' half of the parent tRNA with a steep decline around the anticodon region. However, in the interest of clarity, we have replotted this graph as a heat map of all 468 tRNA loci and have provided this in the figure (Fig. 1D') as per reviewer's suggestion.

We apologise for not referring to the figure for polysome analysis. We have added the citation (now Figure S4C and S4C') in the revised manuscript.

4: Fig. 3 - The use of ASOs to block putative tsRNA interactions is a significant experiment in establishing a function for tsRNA, but is very minimally described. Which ASOs were used to block tsRNAs? Was this only anti-tsGlnCTG - or a panel of ASOs? If the former - are the authors proposing that this is the sole tsRNA responsible for the proliferation regulation? If the latter, what fraction of the tsRNA population was affected?

We thank the reviewer for finding our experiments with ASOs as "a significant experiment in establishing a function for tsRNA".

We apologise for the lack of clarity. We have used a panel of ASOs for this experiment. These include ASOs against tsGlnCTG, tsGlyGCC, tsGlyCCC, tsGluTTC, tsLysTTT which constitute the majority population of tsRNAs observed from the sequencing data. To determine the functioning of ASOs, we used tsGlnCTG level as a proxy. As observed in Figure S3E, there is a decrease in the levels of GlnCTG upon ASO treatment.

We have now mentioned the ASOs used in the results section of the revised manuscript.

“Conversely, blocking 5'-tsRNA function using a pool antisense oligos (against tsGlnCTG, tsGlyGCC, tsGlyCCC, tsGluTTC, tsLysTTT) against tsRNAs during RA-induced differentiation (Fig. S3D) led to a modest increase in AP activity (Fig. 3A).” (Page no. 6)

5: Fig 4B - The relevance of ribosomal proteins found in this analysis is unclear. The authors report that tRFs are associated with translating ribosomes using polysome profiling. Most of the proteins reported in this figure are ribosomal proteins, but some are described as connected to proliferation whereas other to differentiation. It does not appear that ribosome heterogeneity can explain this result. Other explanations should be considered (interaction site?) or a higher stringency of statistical test could be used to eliminate most of ribosomal proteins. Maybe a SILAC based method could be implemented.

As the reviewer rightly pointed out, most of the proteins are ribosomal proteins, alluding to a strong role for tsRNA in translation regulation. In agreement with the reviewer we do believe that one of the possible explanations for enriching differentially interacting ribosomal proteins could be due to the site at which the tsRNA interact. We looked at the positioning of the enriched ribosomal protein on the ribosome and were unable to narrow it down to a specific region (Please refer to the images below). However, with the emerging idea of specialised ribosomes, one cannot rule out the possibility that tsRNAs could exhibit increase associations with a specific set of ribosomes due to the presence of certain ribosomal proteins. Some of the enriched ribosomal proteins such as Rpl10a has been shown to be heterogeneous and provide specificity in translation (Shi *et al.*, 2017). Probing these interactions and potential effects would require a more detailed structural and functional study which we are currently investigating in our follow up study.

This has been described in the discussion section of the revised manuscript

“Additionally, the association of 5'-tsRNAs with different sets of ribosomal proteins between stem versus differentiating cells invoke two exciting possibilities with respect to tsRNA function. First, 5'-tsRNAs may confer functional heterogeneity to ribosomes by associating with specific pools of ribosomal proteins that are involved in the translation of specific targets. Second, the spatial positioning of 5'-tsRNAs on the ribosome, evident from its interaction with both large and small subunit, suggests different modes of translational regulation.” (page no. 11)

6: The results are discussed to large extent within the "Result" section whereas the Discussion covers elements not covered by the MS, including tRF biogenesis and regulation.

We have now extensively reworked the Results and the Discussion sections of the manuscript.

Minor points:

7: One Sentence Summary - if this is intended as running title then should be changed

Apologies for the confusion. We did not intend this to be used as the running title. We have now added a running title "tRNA fragments define cell states"

8: Two different abbreviations are used: tsRNAs and tRFs - could this be changed? Are 5'-tRFs and 5'-tsRNAs equivalent?

For the sake of clarity, we now refer to all the small RNA species derived from tRNAs as tRNA derived small RNAs and we have used 5'-tsRNAs to represent the small RNAs of 30-35 nt size range that are derived from the 5' half of the tRNA (described in this study). This nomenclature has also been used by other groups (Chen *et al.*, 2016; Sharma *et al.*, 2016).

9: LIF - name of LIF protein is not explained when used for the first time.

We have now added this in the revised manuscript (page 3).

10: Is there a correlation between amount of tRNA halves recovered and abundance of tRNA and/or gene copy number?

This is a very interesting question that the reviewer has raised and we thank them for making us look at our data more carefully to analyze the correlation between tRNA gene copy number and abundance of tsRNAs. It is interesting to note that the tsRNA read abundance may not be a function of the tRNA gene copy number. This suggests a specific selection and processing of certain tRNAs to tsRNAs. We have now added this to the supplementary figure (Fig S1D).

We have added the following to the results section (page 4)

“It is intriguing to note that of the 468 tRNA loci that could potentially be processed into 5'-tsRNAs, only a few are selected for processing (Fig. 1D' and S1D). Additionally, there did not seem to be a correlation between the tRNA gene copy number and the abundance of 5'-tsRNAs (Fig. S1D). As can be seen CysGCA and AlaAGC have a high/comparable copy number, but are not selected for processing into tsRNA. Similarly, MetCAT, that has a copy number comparable to GluTTC (one of the highly represented tsRNA), does not produce tsRNAs. In contrast GlyCCC which has a relatively low copy number produces high abundance of tsRNAs, suggesting that is little or no correlation between the tRNA gene copy number and the abundance of tsRNAs (Fig. S1D).”

11: Fig. 2C - could include tsGluGCC

Apologies, there must have been some misinterpretation/typo in the reviewer's comments. We have provided Northern blots for GlyGCC and GluTTC, there are no tRNAs with GluGCC.

12: For direct identification of tRFs-mRNA interaction some previously published datasets could be used (PARIS, LIGR or CLASH methods)

The published data sets that studied the tRF-mRNA interactions deals with a different species of tRNA derived small RNA (approx 18nt) and has been looked at in a different cell system (HEK293) (Kumar *et al.*, 2014). Comparing our results to these reports may not provide additional information regarding mechanistic underpinnings of the tsRNA-target interaction in our system (ESCs).

13: "...angiogenin. the. Additionally,..." - to be corrected

We have now corrected this in the revised manuscript.

14: "spatial positioning of 5'-tsRNAs on the ribosome" could be easily validated by analysis of the position of found ribosomal proteins on the ribosome structure. See also Major point 5.

We mapped the ribosomal proteins identified on the ribosomes as suggested by the reviewer. The ribosomal proteins observed in each condition are distributed at various locations across the ribosome. An in-depth structural study would be necessary to understand the interaction of tsRNA on the ribosome. We are currently following up. Additionally, please refer to the explanation provided to Major point 5.

Referee #3:

The manuscript submitted by Krishna et al. reports on upregulated tRNA 5'-halves (called 5'-tsRNAs in this submission) in mouse embryonic stem cells (mESC) upon retinoic acid-induced differentiation. The authors provide primarily high-throughput data suggesting a role of a subset of these tRNA 5'-halves in modulating protein biosynthesis as well as mRNA stability of certain key factors such as c-Myc.

The topic of tRNA-fragments and their involvement in regulating gene expression is timely, relevant and interesting. However, the manuscript submitted suffers from some over interpretations and the lack of crucial control experiments.

We thank the reviewer for finding our work on tRNA-fragments timely, relevant and interesting. We hope we have addressed the queries raised by the reviewer satisfactorily.

1) The authors state that they have studied the role of several tRNA 5'-halves by overexpressing them. This is misleading since from the Methods chapter it is clear that they used transfection to introduce synthetic RNA into mESC. Strictly speaking this is not "overexpression". It is furthermore pivotal to directly show the extent of increased tRNA 5'-half levels inside the cell (e.g. by northern blot or qPCR). Without this information the data shown in Fig. S3 cannot be interpreted.

We have now added the QPCR quantitation of a candidate 5'-tsRNA (tsGlnCTG) upon transfection (Figure S3A). It is interesting to note that upregulation of tsRNAs in LIF condition failed to produce any phenotypic changes. We agree that the synthetic tsRNA mimics (that are devoid of base modifications) may not replicate the endogenous tsRNAs. Hence we have now changed the term “overexpression” of tsRNAs to “transfection of tsRNA mimics” in the revised manuscript.

“To understand the effect of 5'-tsRNAs in defining cell states, we performed functional studies by either transfecting tsRNA mimics in LIF conditions or by blocking tsRNA function with antisense oligos (ASOs) in RA-induced differentiating conditions. First, we performed alkaline phosphatase (AP) assays as a measure of pluripotency to study the effect of increased tsRNA mimic expression (Fig. S3A) under stem conditions. Transfection of tsRNA mimics collectively (tsGlnCTG, tsGluTTC, tsLysCTT/TTT) or individually in LIF-treated stem-like growth conditions led to a mild decrease in AP activity (Fig. S3B). Moreover, none of the stemness-related markers changed in response to the increased tsRNA levels under LIF conditions (Fig. S3C). Concurrently, we observed minimal changes in the transcriptome when 5'-tsRNAs mimics were transfected under LIF conditions suggesting a minor role for these small RNAs in the stem state (Fig. S3D)”. (Page no. 5)

2) The data shown by the authors point into two possible directions: (1) the tRNA 5'-halves bind the ribosome and polysomes and modulate translation or (2) tRNA 5'-halves associate with specific mRNAs (likely via RNA-binding proteins) and regulate the stability and/or translatability of these messages. The authors need to more rigorously discuss these two options because they somewhat appear mutual exclusive. The *in vitro* translation assays shown in Suppl. Fig. S4B revealed translation inhibition of both their two reporter transcripts (luciferase and GFP). Thus it seems that the regulatory role of the tRNA 5'-halves on translation is mRNA sequence independent since neither luciferase nor GFP can be genuine targets of tRFs in mouse cells.

Our *in vitro* translation studies suggest that tsRNAs are general translational repressors. Nevertheless it is important to note that the *in vitro* translation assay may not completely recapitulate the mechanism of target selection that is observed *in vivo*. The specificity observed in the *in vivo* system can be attributed either to sequence complementarity of the tsRNA to its target or the association of the tsRNAs with certain RNA binding proteins. Interestingly our data suggest that both mechanisms may be employed by tsRNAs to regulate the translation of specific mRNAs. We have included this the discussion section of the revised manuscript.

“In concordance with these reports, our *in vitro* translation studies support the role of tsRNAs as general translational repressors. However, our *in vivo* biochemical assays revealed specificity in the interaction of 5'-tsRNAs with the transcripts.” (Page No. 10)

“This suggests that in RA- treated conditions there could be additional factors that facilitate specific interactions of tsRNAs with their targets. How might this target specificity be achieved? Our study reveals two possible modalities for target selection; either via direct interactions with target transcripts and/or through its interaction with specific set of proteins such as RBPs and ribosomal proteins.” (Page No. 11)

3) related to point #2: the potential mRNA-specific effect of tRNA 5'-halves on mRNA stability seems to be remarkably analogous to miRNA action. Indeed tRNA fragments have been shown in the past to function as mi/siRNAs. Can the authors indeed exclude the possibility that their tRNA 5'-halves associate with Argonaut and function in an RNAi-like fashion? Since they made pull-downs of the tRNA halves and have analysed the interacting proteins, they should actually have the data. If so this should be more explicitly discussed in the text.

While we cannot exclude target selection through sequence complementarity, our mass spec data, did not reveal any significant interactions with Ago or other proteins involved in miRNA mediated regulation, suggesting that tsRNAs may function through other mechanisms. We have discussed this in the revised manuscript.

“Our study reveals two possible modalities for target selection; either via direct interactions with target transcripts and/or through its interaction with specific set of proteins such as RBPs and ribosomal proteins. Our pull-down data from lysates devoid of proteins revealed that tsRNAs could directly interact with target mRNAs, perhaps through sequence complementarity. However,

bioinformatics analysis failed to identify any canonical miRNA-like “seed” region (2-8 nt) in the tsRNAs. Furthermore, we also failed to observe any associations of tsRNAs with the proteins involved in the miRNA pathway in the mass spectrometry results, suggesting the existence of an as yet unidentified target recognition module exhibited by tsRNAs.” (Page No. 11)

4) also related to #2: the authors show that the tRNA fragments under investigation seem to be located in ribosome and polysome fractions on their density gradients. First, the authors should show the polysome gradient profile in order for the reader to assess the quality of their particle separation. Second, there is recent literature on the regulatory role of ribosome-bound small non-coding RNAs. The authors might want to consider discussing their findings in respect to the current state in the field.

We have now added the polysome profiles (now Fig. S4C) as suggested.

The current understanding of tsRNAs is that it binds to the mRNA “tunnel” of the ribosome impeding the movement of the ribosome and dislodging the mRNA from the ribosome (Gebetsberger *et al.*, 2017). However, our data that tsRNA associate with the individual subunits 40 and 60S as well as monosomes and polysomes (Fig S4C') suggests that there may be additional mechanisms in play. We have discussed this in the manuscript.

“In a recent study conducted in *Halofex volcani*, a 26 nt 5'- tRF derived from tRNA^{Val} was shown to bind to a region on the 70S where the mRNA bound to the small subunit thereby competing with the mRNA leading to inefficient translation initiation and dislodging the mRNA (Gebetsberger *et al.*, 2017). While the ~30 nt tsRNAs identified in our study could potentially function through similar mechanisms, detailed structural studies will be required to better understand the functional relevance of tsRNA associations with ribosomal proteins.” (Page No. 11)

5) Fig. 2C: I strongly suggest replacing the northern blots shown by those from Supplementary Fig. S2A. When studying tRNA-fragments it is always important and reassuring to also see the signal of the full length tRNA molecules on the same blot.

We thank the reviewer for their suggestion and have replaced the Fig 2A with Figure S2C.

6) The authors performed small RNA-seq before and after T4 PNK treatment in order to check the status of the 3'-ends of their tRNA halves (Fig. S1D). They conclude that, based on the obtained reads, Angiogenin might not be the main processing enzyme. However there is a 30% increase of tRNA-fragment reads upon T4 PNK treatment which is reminiscent of Angiogenin involvement. In fact the better and more direct way to ascertain an Angiogenin involvement in tRNA halves production would be RNAi against Angiogenin followed by small RNA-seq.

We agree with the reviewer that there is ~15% increase (52-67%) in tsRNA population upon treatment with T4 PNK, suggesting a minor role for angiogenin in this process. We have now added the percentages to the pie chart (now Fig S1E) for clarity. However when we knocked down Angiogenin using 3 different siRNAs in RA condition, we did not observe any significant change in the 30-35 nt population (as assayed using Bioanalyzer). If Angiogenin is the endonuclease that cleaves tRNAs to tsRNAs, one would have expected a decrease in the 30-35 nt RNAs upon knockdown of Angiogenin (similar to what has been reported earlier). These data suggest a potential role for other endonucleases that may function in the biogenesis of tsRNAs.

7) in general: the observed effects involving tRNA 5' halves (e.g. Fig. 1B, Fig. 3A, Fig 5B, Fig 5F, Fig. S3A, B) are quite modest. Therefore a more careful interpretation of the data and more toned-down conclusions are appropriate.

We thank the reviewer for this comment and for making us think about the relevance of subtle changes in the levels of tsRNAs in cell state transitions. While we found only a 15% increase in the overall levels of tsRNAs (in RA-treated condition compared to LIF), our data shows increased association of tsRNAs with transcripts in RA condition compared to LIF (27 vs 420). This suggests that tsRNAs play more dynamic roles in RA conditions even though their levels may be comparable to LIF.

We have made necessary changes and additions to the discussion; explaining both, the reasons for subtlety and a possible role for tsRNAs in fine-control over transcript levels.

“An interesting question raised by our study is how modest changes in levels of tsRNA expression elicits biological response? While we found only a 15% increase in the overall levels of tsRNAs (in RA-treated condition compared to LIF), our data revealed increased association of tsRNAs with transcripts in RA condition compared to LIF (27 vs 420). This suggests that in RA- treated conditions there could be additional factors that facilitate specific interactions of tsRNAs with their targets.”. (Page No. 11)

With regards to subtlety in the effects of tsRNAs, we have added the following lines in the discussion.

“However, it is interesting to note that the effects of tsRNAs are subtle. We believe that these subtle changes are sufficient to facilitate and/or maintain changes in stem-states. For example, in mouse ESCs, it has been shown that moderate alterations of Oct4 levels could lead to reduced pluripotency resulting in differentiation (Niwa *et al.*, 2000). Similarly, small perturbations in cMyc levels can alter the proliferative status of cells (Shichiri *et al.*, 1993). Such examples emphasize the need for a regulatory program that can sustain critical levels of proteins essential for the maintenance of cell-states. We believe that tsRNAs may be involved in fine-tuning translation and target stability of cell-state associated transcripts, thereby altering their protein expression levels.”. (Page No. 12)

References

Anders, S. and Huber, W. (2010) ‘Differential expression analysis for sequence count data’, *Genome Biology*, 11(10). doi: 10.1186/gb-2010-11-10-r106.

- Chan, P. P. and Lowe, T. M. (2009) 'GtRNAdb: A database of transfer RNA genes detected in genomic sequence', *Nucleic Acids Research*, 37(SUPPL. 1). doi: 10.1093/nar/gkn787.
- Chen, Q. *et al.* (2016) 'Sperm tsRNAs contribute to intergenerational inheritance of an acquired metabolic disorder', *Science*, 351(6271), pp. 397–400. doi: 10.1126/science.aad7977.
- Consortium, I. H. G. S. (2001) 'Initial sequencing and analysis of the human genome.', *Nature*, 409, pp. 860–921. doi: <http://dx.doi.org/10.1038/35057062>.
- Darty, K., Denise, A. and Ponty, Y. (2009) 'VARNA: Interactive drawing and editing of the RNA secondary structure', *Bioinformatics*, 25(15), pp. 1974–1975. doi: 10.1093/bioinformatics/btp250.
- Gautam, V. *et al.* (2016) 'An Efficient LCM-Based Method for Tissue Specific Expression Analysis of Genes and miRNAs', *Scientific Reports*, 6. doi: 10.1038/srep21577.
- Gebetsberger, J. *et al.* (2017) 'A tRNA-derived fragment competes with mRNA for ribosome binding and regulates translation during stress', *RNA Biology*. doi: 10.1080/15476286.2016.1257470.
- Genome, M. and Consortium, S. (2002) 'Initial sequencing and comparative analysis of the mouse genome', *Nature*, 420(December 5), pp. 520–562. doi: 10.1038/nature01262.
- Kumar, P. *et al.* (2014) 'Meta-analysis of tRNA derived RNA fragments reveals that they are evolutionarily conserved and associate with AGO proteins to recognize specific RNA targets', *BMC Medicine*, 12(1). doi: 10.1186/s12915-014-0078-0.
- Langmead, B. *et al.* (2009) 'Ultrafast and memory-efficient alignment of short DNA sequences to the human genome', *Genome biology*, 10(3), p. R25. doi: 10.1186/gb-2009-10-3-r25.
- Marcial-Quino, J. *et al.* (2016) 'Stem-loop RT-qPCR as an efficient tool for the detection and quantification of small RNAs in *Giardia lamblia*', *Genes*, 7(12). doi: 10.3390/genes7120131.
- Martin, M. (2011) 'Cutadapt removes adapter sequences from high-throughput sequencing reads', *EMBnet.journal*, 17(1), p. 10. doi: 10.14806/ej.17.1.200.
- Niwa, H., Miyazaki, J. I. and Smith, A. G. (2000) 'Quantitative expression of Oct-3/4 defines differentiation, dedifferentiation or self-renewal of ES cells', *Nature Genetics*. doi: 10.1038/74199.
- Peng, H. *et al.* (2012) 'A novel class of tRNA-derived small RNAs extremely enriched in mature mouse sperm', *Cell Research*, 22(11), pp. 1609–1612. doi: 10.1038/cr.2012.141.
- R Development Core Team, R. (2011) *R: A Language and Environment for Statistical Computing*, R Foundation for Statistical Computing. doi: 10.1007/978-3-540-74686-7.
- Sharma, U. *et al.* (2016) 'Biogenesis and function of tRNA fragments during sperm maturation and fertilization in mammals.', *Science*, 351(6271), pp. 391–396. doi: 10.1126/science.aad6780.
- Shi, Z. *et al.* (2017) 'Heterogeneous Ribosomes Preferentially Translate Distinct Subpools of mRNAs Genome-wide', *Molecular Cell*, 67(1), p. 71–83.e7. doi: 10.1016/j.molcel.2017.05.021.
- Shichiri, M., Hanson, K. D. and Sedivy, J. M. (1993) 'Effects of c-myc expression on proliferation, quiescence, and the G0 to G1 transition in nontransformed cells', *Cell Growth Differ*, 4(2), pp. 93–104.
- Varkonyi-Gasic, E. and Hellens, R. P. (2011) 'Quantitative stem-loop RT-PCR for detection of microRNAs.', *Methods in molecular biology (Clifton, N.J.)*, 744, pp. 145–157. doi: 10.1007/978-1-61779-123-9_10.
- Wickham, H. (2009) 'Elegant Graphics for Data Analysis', *Media*, 35(July), p. 211. doi: 10.1007/978-0-387-98141-3.

2nd Editorial Decision

2 May 2019

Thank you for submitting your revised manuscript. It has now been seen by all of the original referees. As you can see, referees find that their concerns have been sufficiently addressed and recommend publication. However, before I can accept the manuscript, there are some editorial concerns I need you to address.

- Referees have some remaining concerns that need addressing. In particular, referee #1 requests a comparison of mapping between Salmon and Bowtie on an anticodon and isotype level. Moreover, referee #2 recommends textual editing with the purpose of increasing the focus. Please let me know if you would like to discuss further the remaining minor referee concerns.

Thank you again for giving us to consider your manuscript for EMBO Reports, I look forward to your revision.

REFeree REPORTS

Referee #1:

The authors have addressed all my concerns.

Referee #2:

The authors re-submitted improved manuscript and explained data analysis in more depth. I am satisfied with most of the responses, but some clarification remains necessary.

1: The authors attempt to deal with tRNA mapping using a custom Perl script and perform Salmon quantification to assure that is quantification is correct. Unfortunately, the comparison between mapping with Bowtie and Salmon provided was performed on the basis of percentage of mapped reads. The major difficulty of tRNA analysis is the large numbers of overlapping sequences. Due to this "Percentage of reads mapped" using different aligners is strongly expected to be similar, even when the mapping is completely different. The authors should provide a comparison of mapping between Salmon and Bowtie on an anticodon and isotype level.

2: It would have been better to map reads to the database containing both mature and pre-tRNA sequences. However, I would not make this a condition for acceptance.

Referee #3:

The revised version of the manuscript submitted by Krishna et al. has addressed most of my critical points raised and has thus clearly improved. The obtained data are now more carefully described and discussed. The text is still lengthy and quite hard to digest. If the authors get another chance to work on the manuscript I suggest a more focused approach. The manuscript reports on and describes truly interesting findings that are worth to be published. Yet the proposed molecular model how 5' tRNA halves can do both global translation repression as well as mRNA-specific and therefore selective translation simultaneously remains vague. However, given the importance of the performed experiments and the fact that the tRF field is still in its infancy (especially when it comes to molecular details) I would give the benefit of doubt and would suggest acceptance of this manuscript (maybe after another round of minor revisions to streamline the text).

Minor point:

Concerning the response to point #6 of my initial review:

I am not sure how the authors exactly calculate their data, but for me an increase in the fraction of tRF sequence reads from 52.6% to 67.3% (Supplementary Fig. 1E) represents an increase of 28% rather than the stated 15%.

2nd Revision - authors' response

7 May 2019

Referee #1:

The authors have addressed all my concerns.

We thank the reviewer for their suggestions and considering our manuscript for publication.

Referee #2:

The authors re-submitted improved manuscript and explained data analysis in more depth. I am satisfied with most of the responses, but some clarification remains necessary.

We thank this reviewer for reviewing and improving our manuscript. We are happy that the reviewer finds our responses and changes satisfactory.

1: The authors attempt to deal with tRNA mapping using a custom Perl script and perform Salmon quantification to assure that its quantification is correct. Unfortunately, the comparison between mapping with Bowtie and Salmon provided was performed on the basis of percentage of mapped reads. The major difficulty of tRNA analysis is the large numbers of overlapping sequences. Due to this "Percentage of reads mapped" using different aligners is strongly expected to be similar, even when the mapping is completely different. The authors should provide a comparison of mapping between Salmon and Bowtie on an anticodon and isotype level.

As suggested by the reviewer, we performed the correlation between Bowtie and Salmon aligners by grouping the reads based on anticodon. We observe a very strong correlation between the reads aligned by Bowtie and reads aligned by Salmon (Please refer to the figure below). This validates the reproducibility in data produced by two different aligners. However, it must be noted that the tRNAs grouped by anticodon or isotype level may not necessarily be the best way to analyse tsRNAs. The information of sequence differences between tRNA loci for the same anticodon will be lost by grouping tRNAs by anticodon or by isotypes. To overcome this, in our manuscript we grouped tRNAs based on sequence similarity of the first 50 nt and then mapped reads to these grouped tRNAs.

2: It would have been better to map reads to the database containing both mature and pre-tRNA sequences. However, I would not make this a condition for acceptance.

We have used the mature tRNA database to map the reads. This will provide information of the reads that are derived from tRNA processing. Adding pre-tRNA sequences will allow reads to map to introns, the leader and trailer sequences. We believe that this pool however will still remain negligible as a large proportion of reads already map to the mature tRNA database. As evident from our analysis, we can account for ~85% of the small RNA population and any extra small RNA population will be negligible.

Referee #3:

The revised version of the manuscript submitted by Krishna et al. has addressed most of my critical points raised and has thus clearly improved. The obtained data are now more carefully described and discussed. The text is still lengthy and quite hard to digest. If the authors get another chance to work on the manuscript I suggest a more focused approach. The manuscript reports on and describes truly interesting findings that are worth to be published. Yet the proposed molecular model how 5' tRNA halves can do both global translation repression as well as mRNA-specific and therefore selective translation simultaneously remains vague. However, given the importance of the performed experiments and the fact that the tRF field is still in its infancy (especially when it comes to molecular details) I would give the benefit of doubt and would suggest acceptance of this manuscript (maybe after another round of minor revisions to streamline the text).

We thank the reviewer for their suggestions for improving the manuscript. We also thank the reviewer for suggesting the acceptance of this manuscript.

We have now made textual changes to streamline and focus the discussion of the manuscript. We hope that these new edits are satisfactory.

Minor point:

Concerning the response to point #6 of my initial review:

I am not sure how the authors exactly calculate their data, but for me an increase in the fraction of tRF sequence reads from 52.6% to 67.3% (Supplementary Fig. 1E) represents an increase of 28% rather than the stated 15%.

The 15% represents the increase of the tsRNAs as a function of total small RNA population while the 28% population as stated accurately by the reviewer represents the increase of tsRNAs as a function of total tsRNA population.

Thank you for submitting your revised manuscript. I have now looked at everything and all looks fine. Therefore I am very pleased to accept your manuscript for publication in EMBO Reports.

Corresponding Author Name: Ramanuj Dasgupta

Manuscript Number: EMBOR-2019-47789